# Quantifying and Optimizing Simplicity via Polynomial Representations

**Tianren Zhang** [* 1]   **Xiangxin Li** [* 1]   **Minghao Xiao** [* 1]   **Guanyu Chen** [1]   **Feng Chen** [1]

## Abstract

Deep networks often exhibit a preference for "simple" solutions, and such a simplicity bias is widely believed to play a key role in generalization. Yet a broadly applicable, quantitative measure of simplicity remains elusive. We introduce *polynomial representations* as a distribution-aware, low-dimensional surrogate for neural functions: we approximate a network's predictive behavior along data-dependent interpolation paths using orthogonal polynomial bases, yielding a compact functional representation. We show that the *effective degree* of this representation serves as a practical simplicity metric that is predictive of generalization across tasks and architectures, and consistently outperforms existing generalization proxies such as sharpness. Finally, polynomial representations naturally yield a *differentiable* simplicity regularizer, which consistently improves generalization in image and text classification, fine-tuning contrastive vision–language models, and reinforcement learning.[1]

## 1. Introduction

Deep learning models routinely generalize well despite being heavily over-parameterized. A widely discussed explanation is *simplicity bias*: when many solutions fit the training data, the learning dynamics tend to converge to functions that are "simpler" in an appropriate sense, and such simplicity is conjectured to underpin generalization (Hinton & Van Camp, 1993; Gunasekar et al., 2018b; Pérez et al., 2019; Lyu et al., 2021; Huh et al., 2023). However, turning this intuition into a practical tool faces a fundamental obstacle—what exactly does "simple" mean for a neural function, and how can we quantify it in a way that is useful

for modern deep learning?

A useful simplicity measure for neural networks should ideally satisfy three desiderata: **(i) generality** across tasks and architectures, **(ii) quantifiability** at scale for trained models, and **(iii) optimizability** via (approximate) differentiability. Existing proposals typically satisfy only part of these. Provable implicit-bias characterizations such as max-margin or minimum-norm solutions (Soudry et al., 2018; Gunasekar et al., 2018a) provide clean notions of simplicity, but largely hold in restricted regimes and do not extend directly to deep nonlinear models (Chizat et al., 2019; Zhang et al., 2024). Information-theoretic notions based on compression or description length (Schmidhuber, 1997; Dziugaite & Roy, 2017; Arora et al., 2018; Goldblum et al., 2024) offer universal principles, but are difficult to quantify for neural functions and rarely yield a practical training objective. Geometry- or capacity-based notions such as splines and linear-region counts (Montúfar et al., 2014; Raghu et al., 2017) are conceptually aligned with expressivity and hence complexity, yet are architecture-dependent and hard to estimate at scale. Meanwhile, many popular proxies typically live in parameter space (e.g., norms, parameter Jacobian, and sharpness (Neyshabur et al., 2015; Sokolić et al., 2017; Keskar et al., 2017; Lee et al., 2023)) and can be sensitive to *reparameterization* and implementation details (Nagarajan & Kolter, 2019; Andriushchenko et al., 2023). Consequently, it remains challenging to define a simplicity metric that is simultaneously general, quantifiable, and optimizable.

Motivated by this gap, in this work we take a *function-space* approach: if simplicity is ultimately a property of the learned function rather than its parameterization, then quantifying simplicity directly in function space is inherently more compatible with generality and robustness to reparameterization—and, when coupled with a differentiable estimator, can also support optimization. The main challenge is that for an arbitrary high-dimensional function, defining a meaningful notion of simplicity is nontrivial on its own. Our strategy is therefore to introduce a *surrogate function family* that (i) admits a *natural* and tractable notion of simplicity and (ii) can be estimated reliably from data. This leads us to a natural choice of polynomials, which form a simple yet expressive surrogate family—capable of approximating continuous functions on compact domains (Stone, 1948; Trefethen, 2019)—while admitting a natural notion

---

[*]Equal contribution [1]Department of Automation, Tsinghua University, Beijing, China. Correspondence to: Feng Chen <chenfeng@mail.tsinghua.edu.cn>.

*Proceedings of the 43rd International Conference on Machine Learning*, Seoul, South Korea. PMLR 306, 2026. Copyright 2026 by the author(s).

[1]Code is available at: https://github.com/xinzaixinzai/Effective-Degree

of complexity via degree. Concretely, we approximate a network's predictive behavior along data-dependent *interpolation paths* using orthogonal polynomial bases. Paths are formed by interpolating between data points either in the input space (for continuous inputs) or in an appropriate continuous representation space (for discrete inputs), yielding compact, low-dimensional *polynomial representations* that are stable to fit and sidestepping the combinatorial growth of multivariate polynomial bases.

This representation leads to an intuitive notion of simplicity: degree of nonlinearity. Along an interpolation path, low-degree surrogates are close to affine behavior, while higher degrees are needed when predictions vary more nonlinearly. Notably, we show that the *effective degree* of the fitted polynomial expansion provides a general and quantifiable proxy for functional simplicity: across architectures and datasets, it tracks generalization throughout training and is more predictive than widely used measures such as sharpness (Keskar et al., 2017; Kwon et al., 2021).

Beyond measurement, polynomial representations can also lead to simplicity-aware learning objectives. Thanks to the differentiable polynomial fitting procedure, polynomial representations naturally yield a differentiable and tractable simplicity regularizer that explicitly penalizes high-degree components of the learned function. Empirically, we show that incorporating this regularizer into training consistently improves generalization across diverse settings, including image and text classification, fine-tuning contrastive vision–language models, and reinforcement learning.

### Contributions.

1. We propose polynomial representations, a functional surrogate for neural networks obtained by fitting orthogonal polynomials along interpolation paths.

2. We show that the effective degree of this representation provides a simplicity metric that is general and quantifiable across tasks and architectures, tracks generalization, and outperforms existing metrics.

3. We derive a differentiable simplicity regularizer from polynomial representations and demonstrate consistent gains across multiple modalities and tasks.

## 2. Related Work

**Simplicity bias and implicit bias of neural networks.** A large body of work aims to explain generalization in overparameterized networks via implicit regularization from gradient-based training (Neyshabur et al., 2014; Zhang et al., 2017; Belkin et al., 2019). In linear models and certain homogeneous networks, gradient descent provably converges to minimum-norm or max-margin solutions (Soudry et al., 2018; Gunasekar et al., 2018a; Ji & Telgarsky, 2019; Lyu

et al., 2021; Abbe et al., 2023; Zhang et al., 2025), yielding clean simplicity notions in restricted regimes. In deep nonlinear regimes, however, simplicity bias is more elusive: the learned function often depends intricately on architecture, data distribution, and optimization details. Empirically, training dynamics often fit coarse structure before memorizing noise (Arpit et al., 2017; Nakkiran et al., 2019) and exhibit spectral or frequency biases in which low-frequency components are learned earlier or more readily (Rahaman et al., 2019; Xu et al., 2020; Teney et al., 2024). Related function-space analyses use Fourier features, kernels, and linearization (Tancik et al., 2020; Lee et al., 2018; Jacot et al., 2018; Lee et al., 2019), as well as approximation-theoretic function classes such as Barron-type norms (Barron & Klusowski, 2018). Finally, piecewise-linear or spline perspectives exploit the fact that ReLU networks induce spline-like functions with locally simple structure (Montúfar et al., 2014; Raghu et al., 2017; Balestriero et al., 2018), motivating geometric interpretations of representation complexity. While insightful, these lines of work do not directly yield a general, computable simplicity metric. Our work aligns with this function-space view of simplicity bias, but operationalizes it differently by using polynomial representations as a functional surrogate, which naturally admits a general, computable proxy for functional simplicity.

**Generalization proxies for neural networks.** A separate literature proposes post-hoc generalization measures— quantities computed from a trained network that aim to predict or upper bound test error. These include norm- and margin-based measures (Neyshabur et al., 2015; Bartlett et al., 2017), sharpness measures based on local curvature or sensitivity (Keskar et al., 2017; Dinh et al., 2017), as well as PAC-Bayes and compression viewpoints (McAllester, 1999; Dziugaite & Roy, 2017; Arora et al., 2018). However, many of these measures are sensitive to reparameterization or implementation details (Dinh et al., 2017), and their empirical reliability can vary substantially across settings (Jiang et al., 2019). Our metric instead comes from a function-space approximation and is *architecture-agnostic* by construction. Finally, geometric complexity notions such as region counts have also been explored as generalization proxies in classification (Somepalli et al., 2022; Li et al., 2025), but are typically intractable to compute at scale and do not directly provide a differentiable training-time signal.

**Regularization for improving generalization.** Many methods explicitly regularize training to encourage generalization, including weight decay, margin-based penalties (Liu et al., 2016; Elsayed et al., 2018), Jacobian or smoothness regularization (Drucker & Le Cun, 1992; Sokolić et al., 2017; Hoffman et al., 2019; Lee et al., 2023), and sharpness-aware training (Foret et al., 2021; Kwon et al., 2021). In comparison, our regularizer is defined in function space rather than on the model parameters, and is therefore in-

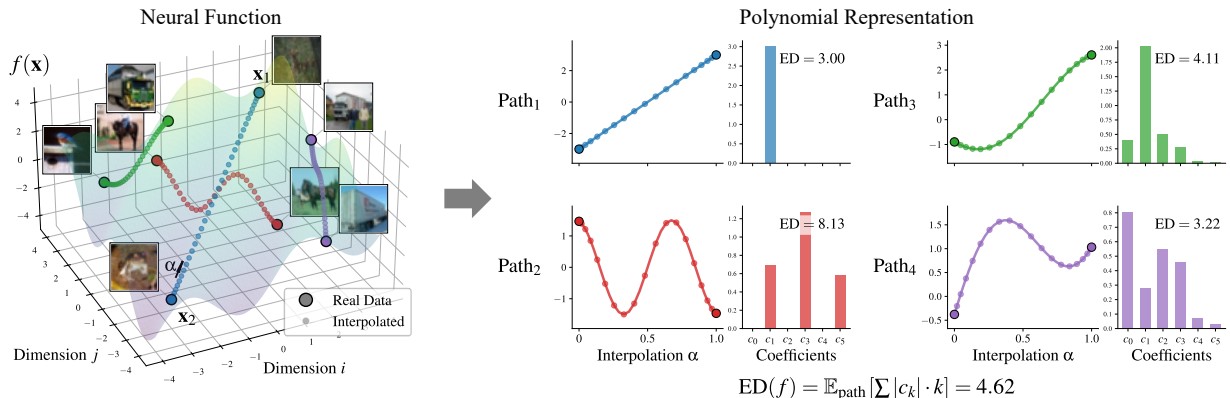

*Figure 1.* **Method overview. Left:** An illustration of the functional landscape of a neural function $f$ by sampling interpolation paths between data points. **Right:** The function $f$'s output along these paths is approximated using polynomial expansions. Coefficient histograms reveal the complexity: smoother paths (e.g., Path$_1$) yield low-degree coefficients, whereas oscillating paths (e.g., Path$_2$) show significant high-degree components. The *effective degree (ED)*, computed by averaging coefficient-weighted degrees of fitted polynomials over multiple interpolation paths, serves as a differentiable proxy for functional simplicity.

variant to reparameterization. Empirically, we compare our method with representative regularizers in this category.

## 3. Polynomial Representations

We estimate functional simplicity by approximating a predictor with a polynomial surrogate and extracting a degree-based notion of complexity. We start from a general polynomial-basis approximation for multivariate inputs, discuss its intractability in high dimensions, and then present key ingredients of our approach: (i) restrict the network to data-dependent one-dimensional interpolation paths, (ii) reduce high-dimensional outputs by PCA, and (iii) ensure numerical stability via orthogonal polynomial bases and a randomized cosine sampling strategy.

**Polynomial basis approximation.** Let $f : \mathbb{R}^d \to \mathbb{R}$ be a (possibly unknown) function. Given a family of polynomial basis functions $\mathcal{B} = \{\phi_{\boldsymbol{\alpha}}(\mathbf{x})\}_{\boldsymbol{\alpha} \in \mathcal{I}}$ indexed by multi-indices $\boldsymbol{\alpha}$ (e.g., monomials), we consider the truncated expansion

$$f(\mathbf{x}) \approx \sum\nolimits_{\boldsymbol{\alpha} \in \mathcal{I}_K} c_{\boldsymbol{\alpha}} \, \phi_{\boldsymbol{\alpha}}(\mathbf{x}), \tag{1}$$

where $\mathcal{I}_K$ collects basis elements up to a prescribed maximum degree $K$. The coefficients $\{c_{\boldsymbol{\alpha}}\}$ can be obtained by least-squares fitting from point evaluations of $f$.

However, directly fitting Equation (1) in the ambient input space is prohibitive when $d$ is large: the number of basis functions in $\mathcal{I}_K$ grows combinatorially with $d$ and $K$, making both fitting and coefficient-based complexity estimation intractable. Moreover, modern predictors may also have high-dimensional outputs (e.g., many logits or token-level outputs), which further increases the computational burden if one were to fit a separate surrogate per coordinate.

**Input reduction.** To obtain a tractable estimate of functional simplicity, we restrict $f$ to data-dependent one-

dimensional paths. Concretely, given $\mathbf{x}_1, \mathbf{x}_2$ drawn from a data distribution $\mathcal{D}$ on $\mathbb{R}^d$, we define the interpolation path

$$\mathbf{x}(\alpha) = \alpha \mathbf{x}_1 + (1 - \alpha)\mathbf{x}_2, \quad \alpha \in [0, 1], \tag{2}$$

and the corresponding univariate restriction

$$g_{\mathbf{x}_1, \mathbf{x}_2}(\alpha) := f(\mathbf{x}(\alpha)). \tag{3}$$

We then fit a *univariate* polynomial surrogate $P_{\mathbf{x}_1, \mathbf{x}_2}(\alpha)$ to $g_{\mathbf{x}_1, \mathbf{x}_2}(\alpha)$ and read off complexity from its coefficients. Averaging over many random pairs yields an estimate anchored to the data distribution while avoiding multivariate polynomial growth. Yet, a natural question is whether restricting to interpolation paths discards the very notion of polynomial complexity/simplicity we aim to measure. In particular, *algebraic degree* is a natural complexity proxy for polynomials, but after reducing the input to one-dimensional interpolations, the resulting univariate restriction could in principle have a smaller degree due to cancellations. Fortunately, we can show that for multivariate polynomials, such degree drops occur only on a measure-zero set of interpolation directions; hence averaging over random interpolation paths preserves the degree ordering almost surely.

**Theorem 3.1** (Order preservation of degree via interpolation paths). *Let $P_1, P_2 : \mathbb{R}^d \to \mathbb{R}$ be nonzero polynomials with degrees $D_i = \deg(P_i)$. Let $\mathbf{x}_1, \mathbf{x}_2 \overset{\text{i.i.d.}}{\sim} \mathcal{D}$ and assume that there exists a nonempty open set $U \subset \mathbb{R}^d$ such that $P(\mathbf{x} \in U) = 1$ and $\mathcal{D}$ admits a density on $U$. Define*

$$d_{P_i}(\mathbf{x}_1, \mathbf{x}_2) = \deg_{\alpha} P_i\big(\alpha \mathbf{x}_1 + (1 - \alpha)\mathbf{x}_2\big).$$

*Then for i.i.d. pairs $(\mathbf{x}_1^{(j)}, \mathbf{x}_2^{(j)})_{j=1}^n$, the empirical averages*

$$\widehat{d}_n(P_i) = \frac{1}{n} \sum\nolimits_{j=1}^n d_{P_i}\left(\mathbf{x}_1^{(j)}, \mathbf{x}_2^{(j)}\right)$$

*satisfy: if $D_1 > D_2$, then $\widehat{d}_n(P_1) > \widehat{d}_n(P_2)$ for all sufficiently large $n$ almost surely.*

*Proof sketch.* This result follows from Lemma A.1, which states that under these assumptions, the interpolation direction $\mathbf{x}_1 - \mathbf{x}_2$ avoids the Lebesgue-null zero set of the leading homogeneous part $P_D$ almost surely, and the univariate restriction $\alpha \mapsto P(\alpha \mathbf{x}_1 + (1 - \alpha)\mathbf{x}_2)$ preserves degree. The key ingredient is the classical fact that the zero set of a nonzero polynomial has Lebesgue measure zero (Basu et al., 2006; Mityagin, 2015). See Appendix A.1 for the full proof. $\square$

**Effective degree.** While Theorem 3.1 justifies path-based dimensionality reduction from the perspective of algebraic degree, directly using it as a practical complexity metric is often brittle: it is sensitive to small coefficient perturbations and may be dominated by negligible high-order components. Motivated by this, we define our simplicity metric with a coefficient-weighted surrogate, which we call the *effective degree*. Concretely, let a fitted univariate polynomial surrogate be represented in its chosen polynomial basis as

$$P(\alpha) = \sum_{k=0}^{K} c_k \phi_k(\alpha), \qquad (4)$$

where $\{\phi_k\}_{k=0}^{K}$ is the basis. We define the effective degree in terms of the fitted coefficients $\{c_k\}$.

**Definition 3.2** (Effective degree). Given Equation (4), the *effective degree (ED)* of $P$ is

$$\mathrm{ED}(P) := \sum_{k=0}^{K} |c_k| \cdot k, \qquad (5)$$

and the normalized version is

$$\mathrm{ED}_{\mathrm{norm}}(P) := \frac{\sum_{k=0}^{K} |c_k| \, k}{\sum_{k=0}^{K} |c_k|}. \qquad (6)$$

For vector-valued surrogates $\mathbf{P} = (P_1, \ldots, P_m)$, we set $\mathrm{ED}(\mathbf{P}) = \frac{1}{m} \sum_{j=1}^{m} \mathrm{ED}(P_j)$ (and similarly for $\mathrm{ED}_{\mathrm{norm}}$). Unlike algebraic degree, ED is Lipschitz in the fitted coefficients, hence more robust to fitting noise. Note that the specific choice of absolute-value weighting here is not unique; we also tested alternative variants such as quadratic weighting and found that the current definition performs best empirically. We posit that this is due to the absolute-value weighting naturally introducing a desirable scale-invariance of the gradient with respect to coefficient magnitude.

**Output reduction.** When the output of $f$ is high-dimensional, fitting a separate surrogate for each coordinate can be costly and statistically noisy. We therefore employ a generic, optional output-compression step before polynomial fitting: we learn a low-dimensional linear subspace that captures most output variation on the sampled path points, and fit polynomials only in this subspace. In particular, we apply PCA (Hotelling, 1933) to the outputs sampled along

each specific interpolation path, retain the top $m$ components, and fit polynomials to the resulting projected outputs, resulting in a *per-path* local dimensionality reduction.

**Numerical stability.** To make coefficient estimation numerically stable, we choose an *orthogonal polynomial basis*. Orthogonality improves conditioning and reduces coefficient instability (Trefethen, 2019). In particular, we use Chebyshev polynomials, defined recursively by $T_0(x) = 1$, $T_1(x) = x$, $T_{k+1}(x) = 2xT_k(x) - T_{k-1}(x)$, and we fit expansions of the form $P(\alpha) \approx \sum_{k=0}^{K} c_k T_k(2\alpha - 1)$. Chebyshev fitting is typically paired with sampling at (shifted) Chebyshev nodes on $[0, 1]$:

$$\alpha_i = \frac{1}{2}\left(1 - \cos \frac{(2i - 1)\pi}{2r}\right), \quad i = 1, \ldots, r, \quad (7)$$

which clusters points near the endpoints and mitigates Runge-type instabilities. In addition, we propose a randomized variant that retains this geometry while providing stratified randomness, which is helpful when $r$ is small and we average over many paths.

**Definition 3.3** (Randomized cosine sampling). Let $r$ be the sampling resolution. We draw

$$\theta_i \sim U\left[\frac{(i - 1)\pi}{r}, \frac{i\pi}{r}\right], \quad i = 1, \ldots, r, \qquad (8)$$

and map to $[0, 1]$ by $\alpha_i = \frac{1}{2}\left(1 - \cos \theta_i\right)$.

Randomized cosine sampling can be viewed as a stratified version of sampling according to the Chebyshev measure, which is known to improve the stability of polynomial least-squares fitting (Cohen et al., 2013).

**Estimator.** Let $P_{\mathbf{x}_1, \mathbf{x}_2}$ denote the fitted degree-$K$ univariate polynomial surrogate to $g_{\mathbf{x}_1, \mathbf{x}_2}(\alpha) = f(\mathbf{x}(\alpha))$ using a sampling scheme and the chosen orthogonal basis. We estimate network simplicity by averaging effective degrees over random interpolation paths:

$$\widehat{\mathrm{ED}}(f) = \mathbb{E}_{\mathbf{x}_1, \mathbf{x}_2 \sim \mathcal{D}}\left[\mathrm{ED}(P_{\mathbf{x}_1, \mathbf{x}_2})\right]. \qquad (9)$$

In practice, we approximate these expectations with finite samples of pairs and path points.

## 4. Effective Degree Tracks Generalization

In this section, we show that effective degree (ED) tracks generalization performance and compares favorably with existing generalization proxies. Specifically, we observe a strong correlation between ED and generalization gap across image classification benchmarks, and find that it accurately captures phase transitions in grokking. Finally, we discuss the relation between ED and standard generalization bounds.

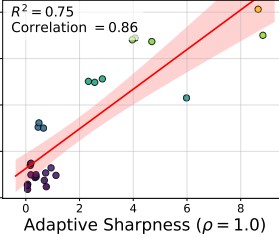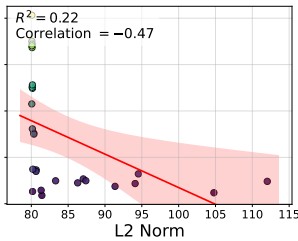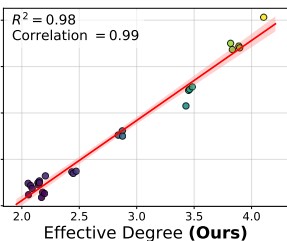

*Figure 2.* Correlations between effective degree, sharpness-based measures, and parameter $L_2$ norm with generalization gap for ResNet18 on CIFAR-10. Effective degree exhibits the strongest linear correlation. Points with lighter colors represent models with larger generalization gaps (same for other figures). Solid red lines indicate least-squares linear fits with 95% confidence intervals.

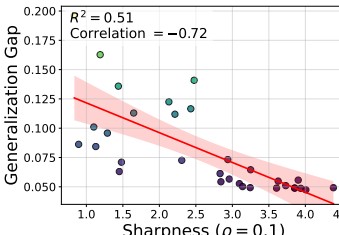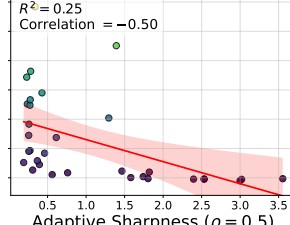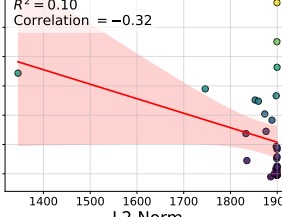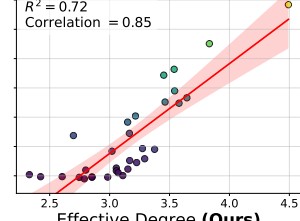

*Figure 3.* Correlations between effective degree, sharpness-based measures, and parameter $L_2$ norm with generalization gap for CLIP ViT-B/32 fine-tuned on ImageNet. Effective degree exhibits a positive correlation with the generalization gap, whereas all other measures correlate negatively. Solid red lines indicate least-squares linear fits with 95% confidence intervals.

### 4.1. ED Correlates with Generalization Gap

We study Pearson correlation between ED and generalization gap (train minus test accuracy) across different training configurations for ResNets (He et al., 2016) / ViTs (Dosovitskiy et al., 2021) on CIFAR-10 (Krizhevsky & Hinton, 2009) and fine-tuned CLIP (Radford et al., 2021) on ImageNet (Deng et al., 2009). We compare ED to sharpness (Foret et al., 2021), adaptive sharpness (Kwon et al., 2021), and parameter $L_2$ norm, reporting the best correlation over metric variants (raw vs. normalized ED; radius sweeps for sharpness). See Appendix D.1 for details.

**CIFAR-10.** We use ResNet18 and ViT-Tiny as representative neural network architectures. For each architecture, we train models across 27 distinct hyperparameter configurations and report results averaged over three random seeds for each configuration. As illustrated in Figure 2, on ResNet18, ED shows the strongest correlation with the gap and consistently outperforms both sharpness variants, while the parameter $L_2$ norm correlates negatively. ViT-Tiny results show similar trends and are detailed in Appendix B.1.

**ImageNet.** We analyze fine-tuned CLIP ViT-B/32 models from Wortsman et al. (2022a) under multiple recipes. Because the fine-tuning recipe such as mixup (Zhang et al., 2018) systematically shifts both the generalization gap distribution and the scale of metrics, we report correlations in a recipe-stratified manner to avoid confounding. Under mixup (Figure 3), ED correlates positively with the gap, whereas sharpness-based measures correlate *negatively*, consistent with prior work (Andriushchenko et al., 2023; da Silva et al.,

2025); the parameter norm is only weakly correlated. Non-mixup recipes show the same qualitative relationship between ED and generalization gap; see Appendix B.2.

### 4.2. ED Tracks Grokking Transitions

We also study grokking (Power et al., 2022) on Modular Division over $\mathbb{Z}_{97}$ with a 30% training split, where models first memorize and only later transition to a generalizing solution. We investigate whether ED can capture this delayed transition compared to established generalization proxies. See Appendix D.2 for detailed experimental protocols.

**ED aligns with the grokking transition.** We track ED alongside parameter $L_2$ norm, sharpness, and adaptive sharpness (best over $\rho$, shown with $\rho = 0.05$). As shown in Figure 4, ED rises during memorization, peaks near the drop in validation loss, and then decreases, suggesting the eventual generalizing solution is simpler under our metric. In contrast, the parameter norm increases monotonically and sharpness measures either decay early or fluctuate, providing a less clear transition signal.

### 4.3. Connecting ED to Generalization Bounds

A classical sanity check for degree-based simplicity is that low-degree polynomials form a low-capacity hypothesis class in standard learning theory. For multivariate polynomials of total degree at most $K$ in $d$ variables, the number of monomials is $M = \binom{d+K}{K}$, and the resulting class can be viewed as a linear predictor in an $M$-dimensional monomial feature space, yielding standard uniform-convergence

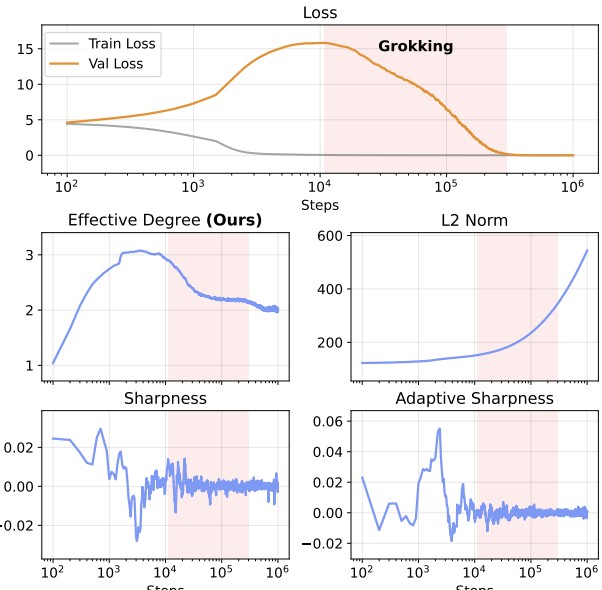

*Figure 4.* Tracking grokking dynamics. **Top panel:** validation loss. **Bottom four panels:** effective degree versus baselines; only effective degree peaks at the transition and decreases thereafter.

generalization guarantees whose dependence worsens as $K$ grows; see, e.g., (Shalev-Shwartz & Ben-David, 2014). Notably, ED resembles a weighted $\ell_1$ constraint on polynomial coefficients; for linear predictors with bounded $\ell_1$ norm and bounded features, classical Rademacher complexity bounds scale only logarithmically with the feature dimension, suggesting that concentrating coefficient mass on lower degrees, as encouraged by smaller ED, corresponds to tighter capacity control (Bartlett & Mendelson, 2002). However, classical uniform-convergence and Rademacher bounds have been known to be *vacuous* for deep neural networks (Dziugaite & Roy, 2017; Nagarajan & Kolter, 2019), for which we show that ED still serves as an effective generalization proxy.

For special model classes such as polynomial neural networks (PNNs), the connection between ED and existing proxies for simplicity or generalization could be made clearer. In particular, we include a controlled study on PNNs in Appendix I, where effective degree preserves the ground-truth algebraic-degree ordering of polynomial target functions, which remains stable under basis changes and PCA-based output reduction.

## 5. Effective Degree Regularization

In this section, we show that effective degree (ED) admits analytic gradients through the polynomial-fitting procedure, enabling end-to-end optimization of functional simplicity as an explicit regularizer.

### 5.1. Differentiability of ED and Stable Implementation

To use ED as a training-time regularizer, we need to differentiate it with respect to network outputs (and hence parameters) through the polynomial fitting step. Concretely, along each sampled interpolation path we fit Chebyshev coefficients by least squares and compute ED as a weighted $\ell_1$ function of these coefficients. Both steps admit closed-form gradients. The following proposition provides the resulting gradient with respect to the sampled outputs.

**Proposition 5.1** (Differentiability of effective degree). *Let $\mathbf{y} \in \mathbb{R}^r$ denote model outputs sampled at $r$ nodes $\{\alpha_i\}_{i=1}^r$ along an interpolation path, and let $\mathbf{c} = [c_0, \dots, c_K]^\top$ be the coefficients of the degree-$K$ Chebyshev least-squares fit. Define $\mathbf{d} = [0, \dots, K]^\top$ and the design matrix $\mathbf{T} \in \mathbb{R}^{r \times (K+1)}$ by*

$$\mathbf{T}_{i,k} = T_k(2\alpha_i - 1), \ i = 1, \dots, r, \ k = 0, \dots, K,$$

*where $T_k$ is the $k$-th Chebyshev basis function. Then, whenever $\mathbf{T}^\top \mathbf{T}$ is invertible and $c_k \neq 0$ for all $k$,*

$$\frac{\partial \text{ED}}{\partial \mathbf{y}} = \mathbf{T}(\mathbf{T}^\top \mathbf{T})^{-1}(\text{sign}(\mathbf{c}) \odot \mathbf{d}), \quad (10)$$

*where $\odot$ is the element-wise product.*

The proof is deferred to Appendix A.2.

While Proposition 5.1 provides the analytical gradient, direct computation involves the inversion of the matrix $\mathbf{T}^\top \mathbf{T}$, which can be numerically unstable when $\mathbf{T}$ is ill-conditioned under stochastic sampling. We therefore solve the damped normal equations to obtain the regularized coefficients $\mathbf{c}_\epsilon$:

$$(\mathbf{T}^\top \mathbf{T} + \epsilon \mathbf{I})\mathbf{c}_\epsilon = \mathbf{T}^\top \mathbf{y}, \quad (11)$$

with a small damping factor $\epsilon > 0$. We can then compute $\mathbf{c}_\epsilon$ via a linear solver,

$$\mathbf{c}_\epsilon = \texttt{LinearSolve}(\mathbf{T}^\top \mathbf{T} + \epsilon \mathbf{I}, \ \mathbf{T}^\top \mathbf{y}), \quad (12)$$

which remains differentiable in modern autodiff frameworks (e.g., PyTorch (Paszke et al., 2019)) and avoids explicit matrix inversion. In practice, we use LU-based linear solves provided by PyTorch. The overhead is modest because each fit is only a small per-path linear solve. The explicit gradient derivation for this damped implementation is provided in Appendix A.2.

### 5.2. ED Regularization Objective

We add ED as a minibatch complexity penalty. Given a minibatch $\mathcal{B} = \{(x_b, t_b)\}_{b=1}^B$, we sample $n_p$ input pairs $\{(x_1^{(i)}, x_2^{(i)})\}_{i=1}^{n_p}$ from $\mathcal{B}$ and evaluate the model along $r$ nodes $\{\alpha_\ell\}_{\ell=1}^r$ either from Chebyshev nodes or randomized cosine sampling. For each pair we fit a degree-$K$ Chebyshev

polynomial to the resulting 1D path and compute its ED. When the output dimension is large, we first project outputs to a low-dimensional subspace via PCA before fitting, as described in Section 3. Note that in our implementation, PCA is performed separately for each sampled path. For the outputs collected along a given path, we compute a path-specific PCA projection, project the path samples into the corresponding low-dimensional subspace, and then fit the polynomial surrogate in this reduced space. Gradients are computed starting from the PCA outputs and differentiated through the PCA decomposition process: after obtaining the path-specific projected representation, the ED regularization loss is computed and differentiated based on those projected coordinates. Averaging over sampled pairs then yields the minibatch estimator $\widehat{ED}_{\mathcal{B}}$. We then optimize

$$\mathcal{L}(\theta; \mathcal{B}) = \mathcal{L}_{\text{task}}(\theta; \mathcal{B}) + \lambda \widehat{ED}_{\mathcal{B}}, \qquad (13)$$

where $\lambda > 0$ controls the regularization strength. The full procedure is summarized in Algorithm 1 in Appendix E.

**Label-anchored ED for classification.** For classification problems, cross-entropy encourages rapid deviation from near-constant predictions early in training, which could lead to an optimization conflict with ED. We mitigate this issue with *label-anchored ED*: we replace the model predictions at the two boundary nodes with the corresponding ground-truth labels when fitting the polynomial surrogate. For randomized cosine sampling, we fix the boundary angles to $\theta_1 = 0$ and $\theta_r = \pi$, and sample only the remaining $r - 2$ interior nodes as in Definition 3.3. Note that this anchoring strategy is compatible with the output reduction described earlier; when both are employed, we apply the label replacement to the boundary nodes *before* computing the PCA projection. Unless otherwise mentioned, in supervised classification tasks we will use this variant and still refer to it as ED.

## 6. Empirical Analysis

In this section, we evaluate ED regularization across vision, language, and reinforcement learning. Concretely, we train ViTs from scratch on CIFAR-10 and ImageNet (Section 6.1), fine-tune CLIP on ImageNet (Section 6.2), fine-tune BERT (Devlin et al., 2018) on GLUE tasks (Wang et al., 2018) (Section 6.3), and regularize reinforcement learning agents on Procgen (Cobbe et al., 2020) (Section 6.4). We also include ablation studies on the hyperparameters and several design choices of ED, analyze its failure mode, and provide a computational overhead analysis (Section 6.5).

### 6.1. Image Classification on CIFAR-10 and ImageNet

On CIFAR-10, we train ViT-Tiny under seven strategies: a standard training baseline, ED regularization with and without label anchoring (LA), mixup (Zhang et al., 2018),

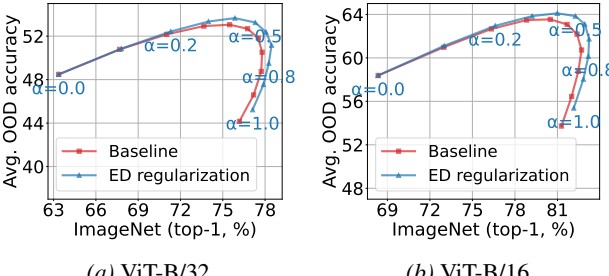

*(a)* ViT-B/32        *(b)* ViT-B/16

*Figure 5.* ImageNet (ID) accuracy vs. average OOD accuracy over 5 shifts under weight interpolation ($\alpha \in [0, 1]$). ED yields a better trade-off than standard fine-tuning across all $\alpha$.

sharpness-aware minimization (SAM) (Foret et al., 2021), ASAM (Kwon et al., 2021), and Jacobian regularization (Hoffman et al., 2019). We also test the scalability of ED regularization by training ViT-S/16 from scratch on ImageNet, following the original ViT recipe (Dosovitskiy et al., 2021) and a stronger recipe from (Beyer et al., 2022) as baselines. See Appendices F.1 and F.2 for details.

**Results.** On CIFAR-10, Table 1 shows that ED yields the best accuracy, improving over the baseline by +3.02 points, while SAM/ASAM/Jacobian reg. are comparable to the baseline. Notably, ED without LA slightly underperforms the standard ED, yet still outperforms all other methods, suggesting that the primary performance gain stems from explicit *complexity control* rather than *label alignment*. On ImageNet, Table 3 shows consistent gains from ED under both recipes, indicating that ED remains effective on larger-scale training and atop carefully tuned recipes.

### 6.2. CLIP Fine-Tuning on ImageNet

We further evaluate the robustness of ED under a family of ImageNet-based distribution-shift benchmarks, including ImageNetV2 (Recht et al., 2019), ImageNet-R (Hendrycks et al., 2021a), ImageNet-A (Hendrycks et al., 2021b), ImageNet Sketch (Wang et al., 2019), and ObjectNet (Barbu et al., 2019). Following Wortsman et al. (2022b), we end-to-end fine-tune CLIP ViT-B/16 and CLIP ViT-B/32 on ImageNet and apply ED regularization during fine-tuning. To incorporate a more advanced fine-tuning recipe, we also follow the weight-space ensembling strategy from Wortsman et al. (2022b) and interpolate between the weights of the zero-shot initialization ($\theta_{\text{ZS}}$) and the fine-tuned model ($\theta_{\text{FT}}$) using a mixing coefficient $\alpha \in [0, 1]$, such that $\theta_{\text{interp}} = (1 - \alpha)\theta_{\text{ZS}} + \alpha\theta_{\text{FT}}$, and evaluate the performance of the interpolated model ($\theta_{\text{interp}}$). Hyperparameters are listed in Appendix F.3.

**Results.** Table 2 summarizes the performance of ED regularization compared to standard fine-tuning. As observed, ED consistently outperforms the baseline across both model architectures. Notably, ED improves performance not only on

*Table 1.* Top-1 test accuracy (%, mean $\pm$ std over 3 seeds) on CIFAR-10. For SAM/ASAM, we report the best result over the tuned $\rho$ grids. For Jacobian regularization, we report the best result over the tuned $\lambda_{\text{JR}}$ grids (see Appendix F.1).

| | Baseline | Mixup | SAM | ASAM | Jacobian reg. | ED w/o LA (Ours) | ED (Ours) |
|---|---|---|---|---|---|---|---|
| ViT-Tiny (Aug) | $87.80 \pm 1.17$ | $88.83 \pm 1.48$ | $87.85 \pm 1.27$ | $87.85 \pm 1.24$ | $87.81 \pm 0.17$ | $90.00 \pm 0.60$ | $\mathbf{90.82 \pm 0.11}$ |

*Table 2.* In-distribution (ID) and out-of-distribution (OOD) accuracy (%, mean $\pm$ std over 3 seeds) for CLIP ViT-B/16 and ViT-B/32.

| Method | ImageNet (ID) | ImageNetV2 | ImageNet-R | ImageNet-A | ImageNet Sketch | ObjectNet | Avg. OOD |
|---|---|---|---|---|---|---|---|
| CLIP ViT-B/32 | $76.20 \pm 0.02$ | $64.21 \pm 0.11$ | $56.82 \pm 0.31$ | $20.48 \pm 0.17$ | $39.08 \pm 0.07$ | $39.62 \pm 0.10$ | $44.04 \pm 0.08$ |
| + ED | $\mathbf{77.14 \pm 0.05}$ | $\mathbf{65.37 \pm 0.18}$ | $\mathbf{58.28 \pm 0.09}$ | $\mathbf{22.03 \pm 0.37}$ | $\mathbf{40.46 \pm 0.24}$ | $\mathbf{40.41 \pm 0.17}$ | $\mathbf{45.31 \pm 0.08}$ |
| CLIP ViT-B/16 | $81.35 \pm 0.11$ | $70.89 \pm 0.13$ | $65.32 \pm 0.10$ | $36.63 \pm 0.11$ | $45.45 \pm 0.32$ | $50.14 \pm 0.14$ | $53.69 \pm 0.04$ |
| + ED | $\mathbf{82.19 \pm 0.03}$ | $\mathbf{72.04 \pm 0.27}$ | $\mathbf{66.30 \pm 0.24}$ | $\mathbf{39.81 \pm 0.66}$ | $\mathbf{47.53 \pm 0.17}$ | $\mathbf{50.76 \pm 0.11}$ | $\mathbf{55.29 \pm 0.14}$ |

*(a) Dodgeball*  *(b) Fruitbot*  *(c) Jumper*  *(d) StarPilot*

*Figure 6.* Generalization on unseen Procgen levels (averaged over 3 seeds). Shaded regions indicate standard errors of the mean.

*Table 3.* Top-1 test accuracy (%) of ViT-S/16 (mean $\pm$ std over 3 seeds) trained from scratch on ImageNet under two recipes.

| Method | Top-1 Accuracy |
|---|---|
| ViT-S/16 (original recipe) | $71.37 \pm 0.17$ |
| + ED | $\mathbf{72.76 \pm 0.16}$ |
| ViT-S/16 (strong recipe) | $74.42 \pm 0.13$ |
| + ED | $\mathbf{75.01 \pm 0.11}$ |

the in-distribution (ID) ImageNet validation set but also on all five *out-of-distribution (OOD)* test sets. These indicate that explicitly enforcing model simplicity enhances robustness against distribution shifts without compromising standard ID generalization capabilities. For the weight-space ensembling setting, Figure 5 plots ImageNet accuracy versus average OOD accuracy across interpolation coefficients. Across the full range, ED-regularized models dominate the baseline trade-off, suggesting improved robustness.

## 6.3. Text Classification on GLUE

We evaluate ED regularization on three GLUE classification tasks (Wang et al., 2018), including two sentence-pair tasks (RTE, MRPC) and one acceptability task (CoLA), using BERT-base (Devlin et al., 2018) and the standard GLUE evaluation metrics. Unlike images, raw-token interpolation is ill-defined. We thus construct interpolation paths in *embedding space* and penalize functional complexity along these paths. Besides the standard BERT fine-tuning baseline, we also compare ED with embedding mixup (Guo et al., 2019), which constructs mixed embeddings for pairs

of examples and trains the classifier to match mixed labels. See Appendix F.4 for training details.

**Results.** Table 4 shows that ED yields improvements across all three tasks, while embedding mixup is not consistently beneficial and can degrade performance. We posit that this is because enforcing *linear* targets on the interpolation path as done by mixup is too stringent for language data. In contrast, ED uses interpolation only to probe and penalize functional complexity rather than to enforce synthetic targets.

## 6.4. Reinforcement Learning on Procgen

Finally, we evaluate ED regularization in reinforcement learning (RL) on the Procgen benchmark (Cobbe et al., 2020), a suite of procedurally generated environments designed to assess generalization capability of RL agents. We train standard CNN-based PPO agents (Schulman et al., 2017) and apply the ED penalty to the actor networks of the agents. Following standard protocols, we train agents on a fixed set of 500 hard-levels (train) and evaluate them on a distinct set of unseen levels (test). We report performance for four environments, including *Dodgeball, Fruitbot, Jumper*, and *StarPilot*. See Appendix F.5 for more details.

**Results.** Figure 6 shows improved generalization across all four environments with ED regularization, with higher asymptotic performance in *Dodgeball, Fruitbot, StarPilot*, and faster learning in *Jumper*. This indicates the applicability of simplicity regularization beyond supervised learning.

*Table 4.* Performance comparison on each task using BERT-base (mean $\pm$ std over 9 seeds).

| Method | RTE (Accuracy) | MRPC (Accuracy) | CoLA (Matthews Corr.) |
|---|---|---|---|
| BERT-base | $70.28 \pm 1.73$ | $86.74 \pm 1.06$ | $62.31 \pm 1.01$ |
| +Mixup | $70.28 \pm 3.07$ | $86.30 \pm 1.22$ | $61.04 \pm 0.91$ |
| +ED | $\mathbf{71.12 \pm 1.96}$ | $\mathbf{87.66 \pm 0.90}$ | $\mathbf{62.45 \pm 1.01}$ |

### 6.5. Ablation, Robustness, and Overhead Analysis

**Protocol.** Implementing ED involves configuring several hyperparameters. To reduce the burden of hyperparameter tuning and demonstrate the robustness of ED, we first use CIFAR-10 as a primary testbed to analyze hyperparameter sensitivity and identify a robust default hyperparameter configuration; for other experiments, we *fix* this unified configuration by tuning only the regularization strength $\lambda$, as described in Appendix F.

We then conduct several ablation studies to better understand the design choices underlying ED.

**Path construction.** To verify the necessity of constructing *data-dependent* interpolation paths, we show in Appendix C.1 that replacing real images with random noise slightly weakens the correlation and leads to poor regularization performance, highlighting the importance of using interpolation paths *near the data manifold*.

We also examine whether ED is restricted to input-space interpolation. As discussed in Section 6.3, for discrete text inputs we already construct paths in embedding space; in Appendix C.3, we additionally evaluate intermediate feature-space interpolation for ViTs on CIFAR-10 and find that ED still improves performance when applied after the embedding layer or after the first Transformer block.

**Polynomial fitting choices.** We examine the choice of polynomial basis in Appendix C.2, showing that replacing the Chebyshev basis with the Legendre basis yields similar performance, suggesting that ED is stable across orthogonal polynomial bases rather than relying on a specific basis choice. Appendix C.4 compares randomized cosine sampling, fixed Chebyshev sampling, and uniform sampling, showing that cosine-based sampling is substantially more stable, especially at higher polynomial degrees.

**Output compression.** Finally, we study whether the optional PCA-based output compression step is responsible for the gains. In Appendix C.5, direct polynomial fitting on the original outputs already works well, and moderate PCA compression to two or three dimensions gives similar performance, suggesting that the gains of ED are not mainly driven by PCA.

**Failure mode and efficiency.** Beyond these controlled ablations, Appendix H provides a failure analysis showing that ED may fail when simpler features are easier to exploit but less desirable for robust generalization. In Appendix G, we provide a brief computational cost analysis and show that although ED regularization increases training time due to the inclusion of additional interpolation examples, the overhead remains acceptable in our experiments.

## 7. Conclusion, Limitations, and Future Work

We introduced effective degree (ED), a general, function-space metric that quantifies the simplicity of a neural network through a polynomial surrogate. Beyond using ED as a post-hoc generalization proxy, we derive analytic gradients through the fitting procedure and develop a numerically stable implementation, enabling ED to be used as an explicit simplicity regularizer during training.

Several directions remain open. On the theory side, it would be valuable to formalize when path-based polynomial surrogates may preserve relevant notions of functional simplicity beyond polynomial degree. On the methodology side, we plan to study alternative bases and sampling schemes, adaptive choices of surrogate degree and resolution, and more efficient estimators that reduce the overhead of regularization at scale. Finally, we will explore ED regularization in additional settings where generalization is brittle, such as long-horizon sequence modeling and out-of-distribution detection, and investigate how ED interacts with pre-training.

## Acknowledgements

This work was supported in part by the National Key Research and Development Program of China No. 2024YDLN0006, and in part by the National Key Research and Development Program of China under STI 2030–Major Projects No. 2021ZD0200300, and in part by the Tsinghua–Fuzhou Data Technology Joint Research Institute (Project No. JIDT2024013).

## Impact Statement

This paper presents work whose goal is to advance the field of machine learning. There are many potential societal consequences of our work, none of which we feel must be specifically highlighted here.

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

# A. Proofs of Theoretical Results

## A.1. Proof of Theorem 3.1

We first prove the following lemma.

**Lemma A.1** (Almost-sure polynomial degree preservation under single interpolation paths). *Let $P : \mathbb{R}^d \to \mathbb{R}$ be a nonzero multivariate polynomial of degree $D \geq 1$. Write its homogeneous decomposition as $P(\mathbf{x}) = \sum_{k=0}^{D} P_k(\mathbf{x})$, where each $P_k$ is homogeneous of degree $k$ and $P_D \not\equiv 0$. Let $\mathbf{x}_1, \mathbf{x}_2 \overset{\text{i.i.d.}}{\sim} \mathcal{D}$, and consider the interpolation path*

$$\mathbf{x}(\alpha) = \alpha \mathbf{x}_1 + (1 - \alpha)\mathbf{x}_2, \qquad \alpha \in [0, 1].$$

*Assume that there exists a nonempty open set $U \subset \mathbb{R}^d$ such that (i) $\mathbb{P}(\mathbf{x} \in U) = 1$ for $\mathbf{x} \sim \mathcal{D}$ and (ii) $\mathcal{D}$ has a density with respect to Lebesgue measure on $U$. Then, with probability one,*

$$\deg_\alpha P(\mathbf{x}(\alpha)) = D. \tag{14}$$

*Equivalently, for the random variable*

$$d_P(\mathbf{x}_1, \mathbf{x}_2) := \deg_\alpha P\big(\alpha \mathbf{x}_1 + (1 - \alpha)\mathbf{x}_2\big),$$

*we have*

$$\mathbb{P}\left(d_P(\mathbf{x}_1, \mathbf{x}_2) = D\right) = 1. \tag{15}$$

*Proof of Lemma A.1.* Fix $\mathbf{x}_2 \in \mathbb{R}^d$ and define the direction $\mathbf{v} = \mathbf{x}_1 - \mathbf{x}_2$. Then the path can be rewritten as $\mathbf{x}(\alpha) = \mathbf{x}_2 + \alpha \mathbf{v}$, and the path restriction is the univariate polynomial

$$q_{\mathbf{x}_2, \mathbf{v}}(\alpha) := P(\mathbf{x}_2 + \alpha \mathbf{v}).$$

We first identify the $\alpha^D$ coefficient of $q_{\mathbf{x}_2, \mathbf{v}}$.

Using the homogeneous decomposition $P = \sum_{k=0}^{D} P_k$, we expand

$$P(\mathbf{x}_2 + \alpha \mathbf{v}) = \sum_{k=0}^{D} P_k(\mathbf{x}_2 + \alpha \mathbf{v}).$$

Since $P_k$ is homogeneous of degree $k$, the highest power of $\alpha$ appearing in $P_k(\mathbf{x}_2 + \alpha \mathbf{v})$ is $\alpha^k$, and its $\alpha^k$ coefficient equals $P_k(\mathbf{v})$. Therefore,

$$P(\mathbf{x}_2 + \alpha \mathbf{v}) = \sum_{k=0}^{D} \Big(\alpha^k P_k(\mathbf{v}) + \text{(lower-order terms in } \alpha)\Big),$$

so the coefficient of $\alpha^D$ in $q_{\mathbf{x}_2, \mathbf{v}}(\alpha)$ is exactly $P_D(\mathbf{v})$. Hence

$$\deg_\alpha P(\mathbf{x}_2 + \alpha \mathbf{v}) = D \quad \Longleftrightarrow \quad P_D(\mathbf{v}) \neq 0.$$

It remains to show $\mathbb{P}(P_D(\mathbf{v}) = 0) = 0$ for $\mathbf{v} = \mathbf{x}_1 - \mathbf{x}_2$. Because $P_D$ is a nonzero polynomial, its zero set $Z := \{\mathbf{v} \in \mathbb{R}^d : P_D(\mathbf{v}) = 0\}$ has Lebesgue measure zero.

Under the assumption that $\mathbf{x}_1, \mathbf{x}_2 \in U$ almost surely and that $\mathcal{D}$ has a density $f$ on $U$, the difference $\mathbf{v} = \mathbf{x}_1 - \mathbf{x}_2$ is absolutely continuous on $U - U$ with density

$$g(\mathbf{v}) = \int_{\mathbb{R}^d} f(\mathbf{u}) \, f(\mathbf{u} - \mathbf{v}) \, d\mathbf{u},$$

(where $f$ is extended by $0$ outside $U$). In particular, $\mathbf{v}$ is absolutely continuous with respect to Lebesgue measure, and thus assigns probability zero to any Lebesgue-null set. Therefore $\mathbb{P}(\mathbf{v} \in Z) = 0$, i.e., $\mathbb{P}(P_D(\mathbf{v}) = 0) = 0$.

Combining the above yields $\deg_\alpha P(\mathbf{x}(\alpha)) = D$ almost surely, proving the claim. $\square$

Now we are ready to prove Theorem 3.1.

*Proof.* By Lemma A.1, for each $i \in \{1, 2\}$ we have $d_{P_i}(\mathbf{x}_1, \mathbf{x}_2) = D_i$ almost surely. Therefore $\mathbb{E}[d_{P_i}] = D_i$, and in particular $D_1 > D_2$ implies $\mathbb{E}[d_{P_1}] > \mathbb{E}[d_{P_2}]$.

For the empirical statement, note that $0 \leq d_{P_i}(\mathbf{x}_1, \mathbf{x}_2) \leq D_i$, so $d_{P_i}$ is integrable. Since the pairs $(\mathbf{x}_1^{(j)}, \mathbf{x}_2^{(j)})$ are i.i.d., the strong law of large numbers gives

$$\widehat{d}_n(P_i) = \frac{1}{n} \sum_{j=1}^{n} d_{P_i}\left(\mathbf{x}_1^{(j)}, \mathbf{x}_2^{(j)}\right) \xrightarrow{\text{a.s.}} \mathbb{E}[d_{P_i}] = D_i.$$

If $D_1 > D_2$, then almost surely there exists $N$ such that for all $n \geq N$, $\widehat{d}_n(P_1) > \widehat{d}_n(P_2)$, because both sequences converge almost surely to distinct limits. $\square$

### A.2. Proof of Proposition 5.1 and Gradients of Damped Least Squares

*Proof.* Our first result regarding the derivative with respect to coefficients,

$$\frac{\partial \text{ED}}{\partial \mathbf{c}} = \text{sign}(\mathbf{c}) \odot \mathbf{d},$$

follows from standard differentiation rules, noting that the subgradient of $|c_k|$ is $\text{sign}(c_k)$ for $c_k \neq 0$.

In the standard polynomial fitting process, we estimate the coefficients $\mathbf{c}$ by solving the ordinary least-squares problem

$$\mathbf{c} = \arg\min_{\mathbf{c}} \|\mathbf{T}\mathbf{c} - \mathbf{y}\|_2^2.$$

When $\mathbf{T}^\top \mathbf{T}$ is invertible, the normal-equation solution is

$$\mathbf{c} = (\mathbf{T}^\top \mathbf{T})^{-1} \mathbf{T}^\top \mathbf{y}.$$

Differentiating both sides with respect to $\mathbf{y}$ yields the Jacobian matrix:

$$\frac{\partial \mathbf{c}}{\partial \mathbf{y}} = (\mathbf{T}^\top \mathbf{T})^{-1} \mathbf{T}^\top.$$

Finally, applying the chain rule yields the gradient presented in Proposition 5.1:

$$\frac{\partial \text{ED}}{\partial \mathbf{y}} = \left(\frac{\partial \mathbf{c}}{\partial \mathbf{y}}\right)^\top \frac{\partial \text{ED}}{\partial \mathbf{c}} = \mathbf{T}(\mathbf{T}^\top \mathbf{T})^{-1} \left(\text{sign}(\mathbf{c}) \odot \mathbf{d}\right).$$

$\square$

**Gradient for the stable implementation (damped least squares).** In Section 5.1, we introduce a damping factor $\epsilon > 0$ to enhance numerical stability. This corresponds to solving the regularized least squares problem (Ridge Regression), where the coefficients $\mathbf{c}_\epsilon$ satisfy:

$$(\mathbf{T}^\top \mathbf{T} + \epsilon \mathbf{I})\mathbf{c}_\epsilon = \mathbf{T}^\top \mathbf{y}.$$

This linear system has the closed-form solution:

$$\mathbf{c}_\epsilon = (\mathbf{T}^\top \mathbf{T} + \epsilon \mathbf{I})^{-1} \mathbf{T}^\top \mathbf{y}. \tag{16}$$

Differentiating both sides with respect to $\mathbf{y}$:

$$\frac{\partial \mathbf{c}_\epsilon}{\partial \mathbf{y}} = (\mathbf{T}^\top \mathbf{T} + \epsilon \mathbf{I})^{-1} \mathbf{T}^\top.$$

Applying the chain rule again, the gradient used in our stable implementation is:

$$\frac{\partial \text{ED}}{\partial \mathbf{y}} = \left(\frac{\partial \mathbf{c}_\epsilon}{\partial \mathbf{y}}\right)^\top \frac{\partial \text{ED}}{\partial \mathbf{c}_\epsilon} = \mathbf{T}(\mathbf{T}^\top \mathbf{T} + \epsilon \mathbf{I})^{-1}(\text{sign}(\mathbf{c}_\epsilon) \odot \mathbf{d}). \tag{17}$$

This confirms that the gradient computation remains valid and analytically tractable when using the damped solver. In the limit $\epsilon \to 0$, Eq. (17) recovers the result in Proposition 5.1.

# B. Additional Correlation Results

This appendix provides supplementary results for the correlation analysis discussed in Section 4.1, including detailed results for ViT-Tiny on CIFAR-10 and CLIP models trained without mixup on ImageNet.

## B.1. ViT-Tiny on CIFAR-10

In the main text, we summarized the correlation analysis for ViT-Tiny. Here, we provide the detailed experimental setup and the corresponding plots.

**Training setup.** We evaluated ViT-Tiny models using a grid of 27 hyperparameter configurations adapted for Transformer architectures. Consistent with the ResNet18 protocol, we report results averaged over three random seeds for each configuration.

**Results.** As shown in Figure 7, effective degree maintains a strong positive correlation with the generalization gap across diverse hyperparameter settings. In contrast, sharpness-based measures exhibit weaker predictive power, while the parameter $L_2$ norm correlates negatively with the generalization gap, failing to capture the correct complexity-generalization relationship in this context.

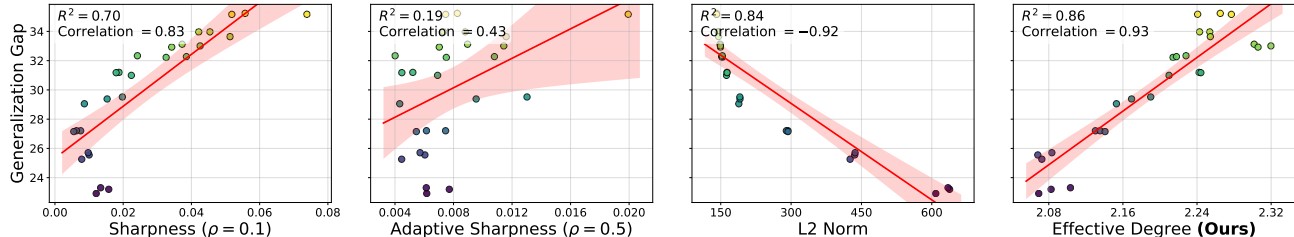

*Figure 7.* Correlation between effective degree, sharpness-based measures, and parameter $L_2$ norm with generalization gap for ViT-Tiny on CIFAR-10. The four panels (left to right) plot generalization gap against standard sharpness, adaptive sharpness, parameter $L_2$ norm, and effective degree, respectively. Each point corresponds to the average over three random seeds under a specific hyperparameter configuration.

## B.2. CLIP Models on ImageNet without Mixup

We also present the analysis on ImageNet to CLIP models fine-tuned *without* mixup augmentation. Figure 8 shows the correlation plots for this setting. Consistent with the mixup-trained models reported in the main text (Figure 3), effective degree maintains a strong positive correlation with the generalization gap. Sharpness-based measures continue to exhibit negative correlations, while the parameter $L_2$ norm shows only a weak relationship with generalization performance.

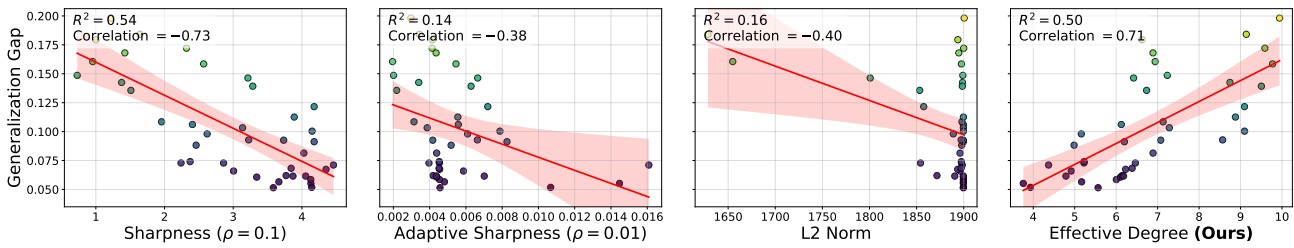

*Figure 8.* Correlation plots for CLIP models trained without mixup on ImageNet. The four panels (left to right) plot generalization gap against standard sharpness, adaptive sharpness, parameter $L_2$ norm, and effective degree, respectively.

*Table 5.* Top-1 test accuracy on CIFAR-10 (%, mean ± std over 3 seeds) for ViT-Tiny under the same explicit regularization setting as in the main text. We compare the Baseline, ED, and an ablation ED (random pixels) that replaces real interpolation endpoints with uniformly sampled random images.

| Model | Baseline | ED | ED (random pixels) |
|---|---|---|---|
| ViT-Tiny | $87.80 \pm 1.17$ | $\mathbf{90.82 \pm 0.11}$ | $87.31 \pm 0.80$ |

## C. Ablation Study

### C.1. Replacing Real Images with Synthetic Random Images in Constructing Interpolation Paths

We study whether the effectiveness of effective degree depends on using *real* data as interpolation endpoints. To this end, we replace the real images of CIFAR-10 used to construct effective degree with i.i.d. sampled uniform noise on each pixel, referred to as random pixels, and evaluate both (i) the correlation between effective degree and the generalization gap and (ii) test accuracy under regularized training. For the correlation analysis, we use ResNet18 following the setup in the main text; for regularized training, we use ViT-Tiny with the same training protocol, hyperparameter configurations, and three-seed averaging as in the main text.

**Correlation.** As shown in Figure 9, replacing real images with random pixels slightly weakens the correlation with the generalization gap on ResNet-18, although the overall positive trend remains.

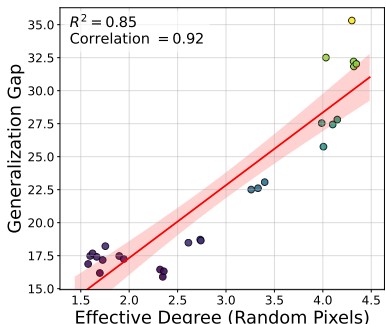

*Figure 9.* Correlation between effective degree computed with uniformly sampled random pixels and the generalization gap on ResNet-18. Points are averaged over three seeds.

**ED regularization.** Table 5 reports the top-1 accuracy on CIFAR-10 for ViT-Tiny under the same explicit regularization setting as in the main text. Notably, ED with random pixels completely removes the gain of ED, indicating that computing ED using on-distribution samples to construct interpolation paths *near the data manifold* is crucial for its effectiveness.

### C.2. Choice of Polynomial Basis

We next examine whether the effectiveness of ED depends on the specific choice of polynomial basis. We use the Chebyshev basis primarily for its numerical stability in polynomial fitting, while viewing the basis as a tractable surrogate for representing the path-restricted function. Since our simplicity notion is degree-based, orthogonal polynomial bases are natural choices. By contrast, non-polynomial bases such as Fourier or wavelet bases would induce related but conceptually different notions of spectral simplicity.

To directly test the dependence on the basis choice, we replace the Chebyshev basis with the Legendre basis on CIFAR-10, while keeping the same hyperparameters under the efficiency-oriented configuration and using three random seeds. The results are reported in Table 6.

The effect of changing the basis is small. This suggests that the method is stable across orthogonal polynomial bases, rather than relying specifically on Chebyshev polynomials.

*Table 6.* Top-1 test accuracy on CIFAR-10 (%, mean $\pm$ std over 3 seeds) for ViT-Tiny with different orthogonal polynomial bases. The small difference between Chebyshev and Legendre bases suggests that ED is stable across orthogonal polynomial bases.

| Basis | Test accuracy |
|---|---|
| Chebyshev | $90.18 \pm 0.14$ |
| Legendre | $89.89 \pm 1.29$ |

## C.3. Linear Blending in Intermediate Feature Space

Our default construction uses input-space interpolation for continuous inputs because it is simple and architecture-agnostic. However, ED is not restricted to input space. For discrete inputs such as text, direct interpolation over raw tokens is not meaningful, and our BERT-based experiments in Section 6.3 therefore construct paths in embedding space. We further investigate whether ED can be applied to intermediate feature spaces for vision models.

Specifically, on CIFAR-10, we apply ED regularization to ViT-Tiny using interpolation paths constructed at three different locations: the input space, the feature space after the embedding layer, and the feature space after the first Transformer attention block. The results are shown in Table 7.

*Table 7.* Top-1 test accuracy on CIFAR-10 (%) under different interpolation spaces. ED still improves performance when applied in intermediate feature spaces, although input-space interpolation gives the largest gain.

| Method | Baseline w/o ED | Input | Embedding | After block 1 |
|---|---|---|---|---|
| Test accuracy | 86.49 | 89.99 | 88.92 | 88.18 |

ED still improves performance in intermediate feature spaces, although the gain is smaller than with input-space interpolation. A plausible reason is that intermediate feature-space interpolation constrains the end-to-end input-output function less directly, even though the regularization signal still backpropagates to earlier layers. That said, we note that our current implementation of feature-space interpolation is preliminary and believe it remains a promising direction to be further explored.

## C.4. Sampling Strategy

We also study the effect of the sampling strategy used to construct points along each interpolation path. We compare three choices: fixed Chebyshev sampling from Equation (7), randomized cosine sampling from Equation (8), and uniform sampling. We evaluate both a low-resolution setting with resolution $r = 4$ and maximum degree $K = 3$, and a higher-degree setting with resolution $r = 15$ and maximum degree $K = 14$. The results are shown in Table 8.

*Table 8.* Top-1 test accuracy on CIFAR-10 (%) under different sampling strategies.

| Resolution $r$ | Max degree $K$ | Sampling | Test accuracy |
|---|---|---|---|
| 4 | 3 | Randomized cosine | 89.99 |
| 4 | 3 | Fixed Chebyshev | 89.47 |
| 4 | 3 | Uniform | 89.23 |
| 15 | 14 | Randomized cosine | 89.90 |
| 15 | 14 | Fixed Chebyshev | 90.40 |
| 15 | 14 | Uniform | 78.42 |

At small resolution and degree, all three sampling strategies perform reasonably well, with randomized cosine sampling giving the best result among the compared settings. In the higher-degree setting, however, uniform sampling becomes unstable and leads to a substantial accuracy drop. In contrast, the two cosine-based strategies remain effective.

## C.5. Dependence on PCA-Based Output Compression

We further study whether the effectiveness of ED depends on the PCA-based output compression step. In our method, PCA is an optional per-path output compression module used mainly for computational efficiency when the model output is

high-dimensional, rather than the source of the ED regularization effect.

To test this, we conduct a CIFAR-10 ablation under the efficiency-oriented setting, comparing direct polynomial fitting on the original model outputs without PCA against PCA projections with different output dimensions. The results are reported in Table 9.

*Table 9.* Top-1 test accuracy on CIFAR-10 (%) under different PCA output dimensions. "No PCA" denotes direct fitting on the original model outputs.

| Output dimension after PCA | No PCA | 3D | 2D | 1D |
|---|---|---|---|---|
| Test accuracy | 89.99 | 89.84 | 89.90 | 88.75 |

These results suggest that ED is not mainly driven by PCA. Direct fitting on the original outputs already achieves strong performance, and PCA compression to two or three dimensions remains very close to the no-PCA result. Only aggressive compression to one dimension noticeably hurts performance, indicating that overly low-dimensional projections may discard useful output variation along the interpolation paths.

## D. Experimental Details for Generalization Analysis

This appendix details the experimental protocols specifically for the correlation and grokking analyses presented in Section 4.

### D.1. Generalization Prediction on CIFAR-10 and ImageNet

We describe the generation of the model pool and the specific protocols for estimating effective degree in the correlation experiments.

**Model pool generation.** To evaluate the correlation between complexity and generalization, we trained a diverse set of models on CIFAR-10 by sweeping over key hyperparameters. For **ResNet18**, following Li et al. (2025), we train models using SGD with a MultiStep learning rate schedule, sweeping over batch sizes $\{256, 512, 1024\}$, learning rates $\{0.1, 0.01, 0.001\}$, and weight decays $\{10^{-5}, 10^{-6}, 10^{-7}\}$. This yields 27 unique configurations, each averaged over three random seeds. For **ViT-Tiny**, we employ the AdamW optimizer (Loshchilov & Hutter, 2019) with cosine learning rate decay, sweeping batch sizes $\{256, 512, 1024\}$, learning rates $\{0.005, 0.001, 0.0005\}$, and weight decays $\{10^{-3}, 10^{-4}, 10^{-5}\}$.

**Baseline configurations.** For sharpness-based baselines, we report the best correlation achieved across a range of neighborhood sizes $\rho$. For standard sharpness, we sweep $\rho \in \{0.01, 0.05, 0.1\}$. For adaptive sharpness, we sweep $\rho \in \{0.01, 0.05, 0.1, 0.5, 1.0\}$. On CIFAR-10, sharpness is computed using the full training set; on ImageNet, we use a subset of 2,048 samples following da Silva et al. (2025).

**Estimation protocol of effective degree.** For the correlation analysis, we employ a high-precision fitting protocol to minimize variance. For both CIFAR-10 and ImageNet experiments, we average results over 400 independent estimations, with the resolution set to 200 and the maximum polynomial degree fixed at 40. We report the best correlation achieved over metric variants (raw vs. normalized ED). Specifically, for the raw variant, we fit the polynomial to the model's output after Softmax (as it provides inherent normalization), whereas for the normalized variant, we fit the logits directly, with normalization explicitly handled within the ED calculation. For ImageNet models (1,000 classes), we additionally use PCA to project outputs onto lower-dimensional subspaces ($m \in \{1, 3, 10\}$) prior to fitting, reporting the highest correlation across these configurations.

### D.2. Grokking Dynamics

We investigate grokking using the Modular Division task over $\mathbb{Z}_{97}$. The model is a 2-layer Transformer trained on a 30% subset of the data using AdamW (learning rate $10^{-3}$, weight decay 0), following the setup of Power et al. (2022).

**Complexity tracking.** We monitor effective degree and baselines throughout the training trajectory. For effective degree estimation during grokking, we sample $n_p = 200$ direction pairs to ensure stability. For the polynomial approximation

---

**Algorithm 1** Training with effective degree (ED) regularization

---

**Input:** Model $f_\theta$; training data $\mathcal{D}$; regularization strength $\lambda$; number of sampled pairs $n_p$; sampling resolution $r$; maximum polynomial degree $K$; jitter $\epsilon$; **optional:** target PCA dim $m$.

**Output:** Trained parameters $\theta$.

 1: **for** each minibatch $\mathcal{B} = \{(\mathbf{x}_b, \mathbf{t}_b)\}_{b=1}^{B}$ **do**
 2:     Compute $\mathcal{L}_{\text{task}}(\theta; \mathcal{B})$.
 3:     $\widehat{\text{ED}}_{\mathcal{B}} \leftarrow 0$.                                        // initialize regularizer estimate
 4:     **for** $p \leftarrow 1$ to $n_p$ **do**
 5:        Sample $(\mathbf{x}_1, \mathbf{t}_1), (\mathbf{x}_2, \mathbf{t}_2)$ uniformly from $\mathcal{B}$.          // require labels $\mathbf{t}_1, \mathbf{t}_2$ to use label-anchoring
 6:        Sample nodes $\{\alpha_\ell\}_{\ell=1}^{r} \subset [0, 1]$ via a sampling scheme.     // Chebyshev nodes or randomized cosine sampling
 7:        Set $\mathbf{x}(\alpha_\ell) \leftarrow \alpha_\ell \mathbf{x}_1 + (1 - \alpha_\ell)\mathbf{x}_2$ for $\ell = 1, \dots, r$.          // linear interpolation between $\mathbf{x}_1$ and $\mathbf{x}_2$
 8:        $\mathbf{y}_\ell \leftarrow f_\theta(\mathbf{x}(\alpha_\ell))$, for $\ell = 1, \dots, r$.
 9:        **If** label-anchored: set $\mathbf{y}_1 \leftarrow \mathbf{t}_2$ and $\mathbf{y}_r \leftarrow \mathbf{t}_1$.        // optional: apply label-anchoring on classification tasks
10:        **If** use PCA: project $\{\mathbf{y}_\ell\}_{\ell=1}^{r}$ to $\mathbb{R}^m$ and set $n \leftarrow m$; **else** set $n \leftarrow \dim(\mathbf{y}_1)$.      // optional: output reduction
11:        **for** $j \leftarrow 1$ to $n$ **do**
12:           $\mathbf{c}^{(j)} \leftarrow \texttt{LinearSolve}(\mathbf{T}^\top\mathbf{T} + \epsilon\mathbf{I}, \mathbf{T}^\top\mathbf{y}^{(j)})$.               // solve coefficients
13:           $\text{ED}_p^{(j)} \leftarrow \sum_{k=0}^{K} |c_k^{(j)}| \, k$.                       // effective degree for $j$-th dim
14:        **end for**
15:        $\text{ED}_p \leftarrow \frac{1}{n} \sum_{j=1}^{n} \text{ED}_p^{(j)}$.                               // average over $n$ dimensions
16:        $\widehat{\text{ED}}_{\mathcal{B}} \leftarrow \widehat{\text{ED}}_{\mathcal{B}} + \text{ED}_p$.
17:     **end for**
18:     $\widehat{\text{ED}}_{\mathcal{B}} \leftarrow \widehat{\text{ED}}_{\mathcal{B}}/n_p$.                                     // average over sampled pairs
19:     $\mathcal{L}(\theta; \mathcal{B}) \leftarrow \mathcal{L}_{\text{task}}(\theta; \mathcal{B}) + \lambda \widehat{\text{ED}}_{\mathcal{B}}$.
20:     Update $\theta$ via gradient descent on $\mathcal{L}(\theta; \mathcal{B})$.
21: **end for**
22: **return** $\theta$.

---

along each 1D slice, we set the resolution to 64 and the maximum degree to 40. This configuration balances computational efficiency with the precision required to track the phase transition.

# E. ED Regularization Pseudocode

We provide the pseudocode for our proposed ED regularization training scheme in Algorithm 1.

# F. Experimental Details for ED Regularization

In this section, we detail the experimental settings for the baselines and the configurations for ED regularization.

To ensure a fair comparison and reproducibility, our experimental settings adhere to the following principles:

- **Baselines:** We strictly follow the hyperparameter configurations reported in the original papers or official open-source implementations. For baselines implemented by us, we perform a standard grid search to ensure optimal performance.

- **ED regularization:** Implementing ED involves configuring several hyperparameters: the regularization strength $\lambda$, sampling resolution $r$, maximum polynomial degree $K$, number of input pairs $n_p$, and the projection dimension $m$ (when using PCA). To reduce the burden of hyperparameter tuning and demonstrate robustness of our method, we first use CIFAR-10 as a primary testbed to analyze hyperparameter sensitivity and identify a robust default **efficiency-oriented configuration** (described in Appendix F.1). We then *fix* these structural parameters except $\lambda$ for all subsequent experiments (ImageNet, language modeling, reinforcement learning). Specifically, we set the sampling resolution to $r = 4$, the maximum polynomial degree to $K = 3$, and the number of sampled pairs to half the batch size ($n_p = B/2$) or the full batch size. For tasks with high-dimensional output spaces (e.g., ImageNet, reinforcement learning), we apply PCA with a projected dimension of $m = 3$ to mitigate computational overhead. Notably, we consistently enforce $m = r - 1$, ensuring that $m$ serves as a dependent variable rather than an additional hyperparameter. Furthermore,

*Table 10.* Top-1 test accuracy on CIFAR-10 (%, mean $\pm$ std over 3 seeds) investigating the performance of the performance-oriented configuration and the efficiency-oriented configuration

| Model | Performance-oriented configuration | Efficiency-oriented configuration |
|---|---|---|
| ViT-Tiny | **90.82 $\pm$ 0.11** | 90.18 $\pm$ 0.14 |

given the resolution $r = 4$, we utilize randomized cosine sampling (Eq. (8)) to enhance the diversity of the sampling points. As detailed in Appendix G, this configuration offers a highly favorable computational profile.

Moreover, we consistently employ the standard (unnormalized) effective degree ED (Definition 3.2) as the regularization penalty. This choice is motivated by the fact that we fit post-softmax probabilities, which are inherently normalized and bounded; thus, the unnormalized ED provides a stable complexity measure without requiring additional scale invariance. Consequently, the only hyperparameter requiring tuning for new tasks is the regularization strength $\lambda$.

### F.1. Settings for CIFAR-10

The baseline, ED regularization, mixup, SAM and ASAM share the same backbone, optimizer, learning-rate schedule, and preprocessing, and differ only in the corresponding training strategy. We train all models using AdamW for 300 epochs, with batch size 256, base learning rate 0.005, and weight decay 0.1. The learning rate is linearly warmed up for the first 10 epochs and then decayed with cosine annealing to a minimum learning rate of 0. Random cropping and horizontal flip data augmentations are used.

**Mixup.** For mixup, we sample the mixing coefficient $\lambda \sim \mathrm{Beta}(\alpha, \alpha)$ with $\alpha = 1.0$, which reduces to a uniform distribution on $[0, 1]$. Given two training examples $(x_1, t_1)$ and $(x_2, t_2)$, mixup constructs a mixed sample $\tilde{x} = \lambda x_1 + (1 - \lambda)x_2$ with the corresponding soft label $\tilde{t} = \lambda t_1 + (1 - \lambda)t_2$.

**SAM/ASAM.** For sharpness-aware minimization (SAM) and its adaptive variant (ASAM), we follow the standard formulations and tune the neighborhood size $\rho$ using the grids from the original papers. Specifically, for SAM we use $\rho \in \{0.01, 0.02, 0.05, 0.1, 0.2, 0.5\}$, and for ASAM we use $\rho \in \{5 \times 10^{-5}, 10^{-4}, 2 \times 10^{-4}, \ldots, 0.5, 1.0, 2.0\}$. For each method, we report the best result over the corresponding $\rho$ grid (selected by top-1 test accuracy).

**Jacobian regularization.** For Jacobian regularization (JR), we employ random projections to efficiently approximate the Jacobian norm. We tune the regularization coefficient $\lambda_{\mathrm{JR}}$ over the grid $\{0.01, 0.05, 0.1, 0.5, 1.0\}$. Experiments are repeated with three random seeds, and we report the best result selected by top-1 test accuracy.

**ED regularization.** For ED regularization, we implement a ramp-up schedule for the regularization strength over the first 100 epochs, increasing sinusoidally from 0 to $\lambda$. This approach facilitates rapid task learning in the early stages of training before enforcing stronger complexity control. Additionally, we adopt the label-anchored ED strategy described in Section 5.2. Regarding the structural hyperparameters, specifically resolution $r$, degree $K$, and the number of pairs $n_p$, we explore two distinct configurations to balance performance and efficiency:

- **Performance-oriented configuration:** In our initial search for optimal performance, we observe that higher sampling resolutions generally improve the stability and effectiveness of the regularizer with a fixed Chebyshev-node sampling scheme (Eq. (7)). The optimal setting is $r = 15$, $K = 7$ with regularization strength $\lambda = 2$ and compute ED using $n_p = 256$ sampled pairs within each minibatch. While these settings yield the best accuracy, they incur higher computational costs.

- **Efficiency-oriented configuration:** To enhance practical applicability, we evaluate a lightweight configuration characterized by $r = 4$, $K = 3$, and $n_p = 128$ (corresponding to half the batch size) with randomized cosine sampling. For this setting, the regularization strength was determined to be $\lambda = 7$ based on empirical search. Despite the reduced resolution, this setting achieves competitive results (as shown in Table 10) while significantly reducing training time. Consequently, this serves as a standard setting where resolution parameters are fixed, reducing the hyperparameter search space to only $\lambda$.

## F.2. Settings for ImageNet

For the ImageNet experiments, we employ the ViT-S/16 architecture and evaluate our method under two training recipes:

- **Original recipe:** This protocol largely follows the standard recipe outlined in Dosovitskiy et al. (2021). To accommodate computational constraints, we adjust the configuration by training for 90 epochs using the AdamW optimizer with a reduced global batch size of 1024. All other hyperparameters remain unchanged.

- **Strong recipe:** We adopt the improved training recipe and hyperparameter settings proposed by Beyer et al. (2022) without mixup augmentation, which serves as a stronger baseline.

**ED regularization.**   We apply the same ED regularization configuration across both training settings. The regularization setup shares similarities with our CIFAR-10 experiments but includes adaptations for the larger output space. We employ the label-anchored ED strategy in conjunction with randomized cosine sampling. To efficiently scale to the 1,000-class output space, we apply PCA to reduce the model outputs to 3 principal components before fitting the polynomials, consistent with the strategy described earlier. We configure the hyperparameters as follows: regularization strength $\lambda = 3$, sampling resolution $r = 4$, maximum polynomial degree $K = 3$, and number of sampled pairs per minibatch $n_p = 512$. Similar to the CIFAR-10 settings, we apply a sinusoidal ramp-up schedule for $\lambda$ during the first 30 epochs (approx. 1/3 of the training duration) to allow the model to learn adequate representations before enforcing stronger complexity control.

## F.3. Settings for CLIP Fine-Tuning on ImageNet

We adhere to the end-to-end fine-tuning protocol outlined in Wortsman et al. (2022b). Specifically, we train the pre-trained models for 10 epochs with a batch size of 512 and a fixed learning rate of $3 \times 10^{-5}$.

**ED regularization.**   We adopt the same label-anchored ED strategy and PCA projection as in our ImageNet experiments from scratch. For this fine-tuning setting, we adjust the hyperparameters to $\lambda = 2$ and $n_p = 256$ (maintaining $r = 4$ and $K = 3$). Notably, we omit the warmup schedule for $\lambda$ to enforce immediate regularization, as required by the short 10-epoch training duration.

## F.4. Settings for Natural Language Processing Tasks

We fine-tune the BERT-base model on the RTE, MRPC, and CoLA datasets. To ensure a rigorous comparison, we first establish strong baselines by performing a grid search over the following hyperparameters: learning rate $\in \{2 \times 10^{-5}, 5 \times 10^{-5}, 1 \times 10^{-4}\}$, batch size $\in \{16, 32, 64\}$, and training epochs $\in \{5, 10, 20\}$. All reported results are averaged over 9 random seeds.

**Adaptation for embedding interpolation.**   Unlike images, raw text data consists of discrete tokens, prohibiting direct linear interpolation in the input space. To enable the construction of interpolation paths for ED regularization, we adopt the *embedding interpolation* strategy inspired by (Guo et al., 2019). Specifically, let $E(\cdot)$ denote the embedding function that maps input tokens to their vector representations (aggregating word embeddings, positional encodings and segment embeddings). Given two sentence inputs $x_1, x_2$, we perform linear interpolation on their embedding representations immediately after encoding: $h(\alpha) = \alpha E(x_1) + (1 - \alpha)E(x_2)$. This intermediate manifold is then propagated through the transformer layers to compute the effective degree of the decision trace.

**Method-specific configurations.**   For mixup, we employ an embedding interpolation strategy $\lambda \sim \text{Beta}(\alpha, \alpha)$ with $\alpha = 1.0$. For ED regularization, we adopt the label-anchored ED strategy with randomized cosine sampling. Across all tasks, we fix the sampling resolution to $r = 4$ and the maximum polynomial degree to $K = 3$. Consistent with the short fine-tuning duration, we do not apply a warmup schedule for regularization strength $\lambda$.

Based on the grid search, the optimal baseline configurations and the task-specific ED hyperparameters (regularization strength $\lambda$ and sampled pairs $n_p$) are set as follows:

- **RTE:** We use a learning rate of $5 \times 10^{-5}$, batch size 32, and train for 10 epochs. For ED, we set $\lambda = 0.5$.

- **MRPC:** We use a learning rate of $5 \times 10^{-5}$, batch size 16, and train for 10 epochs. For ED, we set $\lambda = 0.5$.

- **CoLA:** We use a learning rate of $2 \times 10^{-5}$, batch size 64, and train for 20 epochs. For ED, we set $\lambda = 0.3$.

### F.5. Settings for Reinforcement Learning

Following standard protocols (Wang et al., 2020), we train agents on a fixed set of 500 levels (train) and evaluate them on a distinct set of unlimited unseen levels (test). We adhere to the implementation details and base hyperparameters provided in Huang et al. (2022). Agents are trained for 50M timesteps, with the exception of *FruitBot*, which is trained for 100M timesteps to accommodate its longer startup phase.

**Adaptation for actor regularization.** We apply our method to the Proximal Policy Optimization (PPO) algorithm, utilizing a CNN-based actor-critic architecture. The actor network $\pi_\theta(a|s)$ determines the agent's behavior, while the critic $V_\phi(s)$ estimates the value function. We introduce the ED penalty exclusively to the actor network. The rationale for this design choice is twofold: First, the Actor defines the decision boundary; ensuring the smoothness of the policy with respect to state variations directly translates to more robust and generalized action selection. Second, the value function often requires sharp transitions to accurately reflect sudden changes in expected returns (e.g., cliffs or immediate rewards), making complexity penalization potentially detrimental to value estimation.

**ED regularization.** For the regularization setup, we employ randomized cosine sampling in conjunction with PCA dimensionality reduction. Specifically, we project the output vectors of the policy network onto a subspace of $m = 3$ dimensions. We adopt the same base hyperparameters for all four tasks: regularization strength $\lambda = 0.01$, sampling resolution $r = 4$, maximum polynomial degree $K = 3$ and sampled pairs $n_p = 2048$ (half of the minibatch size). For *Fruitbot*, *Jumper*, and *StarPilot*, we use a sinusoidal ramp-up schedule for the first $1/6$ of the training steps.

Distinct from the supervised learning settings in previous sections, fixed ground-truth action labels are unavailable in reinforcement learning. We thus employ ED regularization without the label-anchoring strategy and directly constrain the effective degree of the policy network to induce a simpler actor.

## G. Computational Efficiency Analysis

In this section, we analyze the computational overhead incurred by the additional sampling process. Our analysis is based on the efficiency-oriented configuration described in Appendix F.1, which was shown to yield competitive results in our experiments. Specifically, this setting employs a sampling resolution of $r = 4$ and a pair count of $n_p = B/2$, where $B$ denotes the batch size.

Under the label-anchored ED strategy, the two endpoints of the interpolation trajectory are fixed to the ground-truth anchors and do not require model forward passes. Consequently, gradient propagation is only required for the $r - 2 = 2$ intermediate nodes per sampled pair. Given that we sample $n_p = B/2$ pairs per iteration, the total number of additional forward passes is:

$$\text{Additional Passes} = n_p \times (r - 2) = \frac{B}{2} \times 2 = B.$$

This overhead is exactly equivalent to the computational cost of processing one nominal batch. Since the polynomial fitting operations are fully parallelized and incur negligible cost compared to the model forward passes, this implies a theoretical cost multiplier of approximately $2\times$.

To validate this empirically, we conducted time-cost experiments on CIFAR-10 with a batch size of $B = 256$. The average time per epoch was 6.14s without regularization, whereas introducing ED regularization increased this to 9.75s. We further verified these observations using the CLIP fine-tuning setup ($B = 512$), where the average time per iteration increased from 0.44s (without regularization) to approximately 0.90s (with ED). These results align with the theoretical prediction of a $2\times$ increase, as the specific GPU resource demands differ between these two tasks.

## H. Failure Analysis

We include a failure case to clarify a limitation of ED regularization. Motivated by prior work on simplicity bias, we hypothesize that ED regularization may fail when the simpler feature is also the less robust one. To test this hypothesis, we construct MNIST–CIFAR following Shah et al. (2020), a synthetic dataset formed by concatenating MNIST images from

*Table 11.* Failure case on MNIST–CIFAR. Both the baseline and ED-regularized model rely almost entirely on the simpler MNIST signal. Randomizing the MNIST component causes accuracy to collapse, whereas randomizing the CIFAR component has almost no effect.

| Metric | Baseline | ED |
|---|---|---|
| Test acc. | 99.85 | 99.90 |
| Test acc. (MNIST randomized) | 48.05 | 47.95 |
| Test acc. (CIFAR randomized) | 99.85 | 99.90 |

classes 0 and 1 with CIFAR-10 images from the automobile and truck classes. The resulting binary classification task can be solved using either simple MNIST features or more complex CIFAR-10 features.

We train ViT-Tiny with standard training and with ED regularization. To assess which component the model relies on, we evaluate standard test accuracy, test accuracy after randomizing the MNIST component, and test accuracy after randomizing the CIFAR-10 component. The results are shown in Table 11.

Both models achieve near-perfect standard test accuracy. However, when the MNIST component is randomized, accuracy drops to nearly chance level, while randomizing the CIFAR-10 component leaves accuracy essentially unchanged. This indicates that both models rely almost entirely on the simpler MNIST signal. ED regularization does not reduce this reliance or improve robustness in this setting.

We therefore view MNIST–CIFAR as a plausible failure case of ED regularization: when simple features are easier to exploit but less desirable for robust generalization, ED regularization may fail to encourage the use of more generalizable complex features.

## I. Controlled Study on Polynomial Neural Networks

We further examine effective degree in a controlled setting where the notion of algebraic degree is intrinsic to the model class. Specifically, we consider polynomial neural networks (PNNs), which provide a simple setting for testing whether effective degree can recover the intended complexity ordering of polynomial target functions. This experiment also complements the discussion in Section 4.3 by connecting effective degree to settings where complexity and generalization are more directly characterized.

We consider learning polynomial mappings from $\mathbf{x} = (x_1, x_2, x_3) \in \mathbb{R}^3$ to $\mathbf{y} = (y_1, y_2, y_3) \in \mathbb{R}^3$. We construct three target mappings with increasing algebraic degree:

$$\text{Task 1:} \quad \mathbf{y} = (x_3 + 2, \ x_2 + 3, \ x_1 + 1),$$
$$\text{Task 2:} \quad \mathbf{y} = (x_1 x_2, \ x_2 x_3, \ x_1 x_3),$$
$$\text{Task 3:} \quad \mathbf{y} = \left(x_1 x_2 x_3, \ x_1^2 x_2 x_3^2, \ x_1^2 x_2 x_3 + x_2^2 x_3\right).$$

Tasks 4–6 are constructed by multiplying the outputs of Tasks 1–3 by a factor of 2, respectively, while preserving their algebraic degrees. This allows us to test the scale dependence of effective degree and its normalized variant.

We train a 4-layer PNN with square activation functions, consisting of three hidden layers followed by a linear output layer. The model is trained to fit each target mapping nearly perfectly. We then evaluate effective degree and several variants, including normalized effective degree, Legendre-basis effective degree, and PCA-based output reductions. The results are summarized in Table 12.

These controlled polynomial-learning experiments reveal several useful properties of effective degree. First, effective degree preserves the degree ordering of the target mappings: Tasks 1, 2, and 3 have increasing algebraic degree, and their corresponding effective degree values increase accordingly. The same ordering also holds for the scaled versions, Tasks 4, 5, and 6. Second, the standard effective degree is not scale-invariant: scaling the outputs by a factor of 2 approximately increases the corresponding effective degree values. In contrast, $\text{ED}_{\text{norm}}$ is nearly invariant to this scaling. Third, the same degree ordering is preserved when replacing the Chebyshev basis with the Legendre basis, suggesting that the method does not rely on a specific orthogonal polynomial basis. Finally, PCA-based output reduction to one or two dimensions still preserves the same ordering, indicating that effective degree remains effective under moderate dimensionality reduction.

*Table 12.* Controlled study on polynomial neural networks. We report ED and its variants on polynomial target mappings with increasing algebraic degree. Tasks 4–6 are scaled versions of Tasks 1–3, respectively.

| Task | Algebraic degree | ED (Chebyshev) | $ED_{norm}$ (Chebyshev) | ED (Legendre) | ED (PCA-1) | ED (PCA-2) |
|------|------------------|----------------|--------------------------|---------------|------------|------------|
| Task 1 | 1 | 0.72 | 0.29 | 0.73 | 1.40 | 0.70 |
| Task 2 | 2 | 1.17 | 0.78 | 1.33 | 2.12 | 1.34 |
| Task 3 | 5 | 2.92 | 1.17 | 3.64 | 5.32 | 3.40 |
| Task 4 | 1 | 1.34 | 0.28 | 1.35 | 2.67 | 1.34 |
| Task 5 | 2 | 2.33 | 0.82 | 2.63 | 4.13 | 2.62 |
| Task 6 | 5 | 5.30 | 1.17 | 6.64 | 9.62 | 6.16 |

