# OpenReview forum: "Quantifying and Optimizing Simplicity via Polynomial Representations"
_ICML.cc/2026/Conference — ICML 2026 regular_

### Official Review · Reviewer_82s4 · 2026-03-07

**Soundness:** 3
**Presentation:** 4
**Significance:** 3
**Originality:** 3
**Overall Recommendation:** 5
**Confidence:** 5

**Summary:**

This paper proposes a simplicity metric in function-space for neural networks based on polynomial representations along interpolation paths.
The core idea is to restrict the network to one-dimensional paths between data points, fit a degree-K polynomial surrogate along each path, and define a complexity metric called effective degree (ED) from the fitted coefficients.
The paper argues that ED serves both as a post-hoc proxy for generalization and as a differentiable training-time regularizer.
Empirically, the authors report that ED correlates strongly with generalization gap and that ED regularization improves performance across image classification, language tasks, CLIP fine-tuning, and reinforcement learning.

**Compliance With Llm Reviewing Policy:**

Affirmed.

**Final Justification:**

I really appreciate the authors' response. Most of my concerns are clarified. The additional controlled experiment with PNNs is insightful.
I enjoy reading this paper. I believe this paper has good contribution to the community. I recommend this paper for acceptance.

**Key Questions For Authors:**

- Theorem 3.1 is stated for exact algebraic degree, whereas the practical method uses effective degree. Can the authors provide any theoretical result that directly supports ED itself, for example a stability, consistency, or approximation guarantee for the fitted ED estimator?
- Since ED is defined from the fitted coefficients, how sensitive is the metric to the choice of polynomial basis? Have the authors tried other orthogonal bases, such as Legendre polynomials, and if so, do the conclusions remain qualitatively the same?
- How should one choose the surrogate degree $K$ and the sampling resolution $r$ in practice? As shown in the appendices, the paper uses different values across experiments. Is there any principled model-selection criterion, or are these choices mainly heuristic?
- For high-dimensional outputs, the method applies PCA separately on each path before fitting the surrogate. How much of the measured ED is driven by this compression step? Can the authors provide an ablation on the PCA dimension $m$ or compare against fitting directly on the original outputs when feasible?
- More broadly, under what conditions do the authors expect path-based polynomial surrogates to preserve meaningful notions of function-space simplicity beyond exact polynomial degree? This seems central to the conceptual interpretation of the method.
- The proposed ED is attractive because it is differentiable, but what is the principled justification for this particular linear weighting? Since the metric depends directly on coefficient magnitudes, how do the authors ensure it captures functional complexity rather than scale effects, especially when outputs/features may have very different norms? Have the authors tested alternative weighting schemes or normalizations?
- Since the cleanest theoretical setting of the paper is the polynomial case, have the authors considered a controlled study on polynomial neural networks with known degree? Such an experiment could test whether the proposed effective degree recovers the intended ordering of complexity, and whether it is robust to coefficient scaling, basis choice, and the additional approximation steps such as path restriction and PCA.

**Limitations:**

yes

**Strengths And Weaknesses:**

**Strengths**

This paper focuses on an important problem and is well-written in general. The authors aim to measure the "simplicity" of trained neural networks in function space through polynomial representation. This is a compelling perspective, and the proposed ED metric is relatively intuitive and computationally tractable compared with many abstract complexity notions.
The methodology is technically sound.
The proposed metric ED can be used both as a measurement tool and a regularizer since it is differentiable.
The empirical section is also strong.
The authors show a strong correlation between ED and the generalization gap.
They further show that ED can track grokking dynamics nicely.
The fact that the regularizer is evaluated across several modalities and settings also makes the paper more convincing than a narrow single-benchmark study.

**Weaknesses**
- The theoretical justification seems somewhat disconnected from the practical metric. Theorem 3.1 is about preservation of exact algebraic degree under random interpolation paths, showing that degree drops occur only on a measure-zero set and that degree ordering is preserved asymptotically. However, the practical method does not use exact degree; it uses the coefficient-weighted effective degree because exact degree is brittle and sensitive to small perturbations. As a result, Theorem 3.1 feels more like a sanity check for the path-restriction idea than a strong justification of the actual metric used in the experiments.
- The paper defines ED directly from the fitted coefficients $a_k$ so the metric depends on the chosen polynomial basis. The authors choose an orthogonal basis for numerical stability and specifically use Chebyshev polynomials with Chebyshev-type sampling for conditioning and stability reasons. This is reasonable numerically, but it raises the question of whether the measured simplicity is intrinsic or partly an artifact of the chosen representation. The paper itself lists “alternative bases and sampling schemes” as future work. I think further study in this direction will greatly increase the impact of this paper.
- Several key hyperparameters appear to be chosen heuristically rather than from a principled rule. The surrogate depends on the maximum degree $K$, the path sampling resolution $r$, the number of sampled pairs, and, when outputs are high-dimensional, the PCA dimension $m$.The paper acknowledges the need for “adaptive choices of surrogate degree and resolution” as future work. Since ED itself depends on these choices, I think the methodology would be stronger with either a principled selection rule or a more systematic sensitivity analysis.
- While the effective degree is appealing because it is differentiable and can therefore be used as a regularizer, the paper does not seem to sufficiently justify the specific weighting scheme $ED(P) := \sum_k |a_k| k$. The current motivation appears to be mainly robustness relative to exact algebraic degree, but it is not clear why linear weights in $k$ are the right choice, as opposed to other monotone or normalized alternatives. More importantly, since ED depends directly on coefficient magnitudes, the metric may conflate polynomial order with amplitude/scale effects, especially when different outputs or features have different norms, and when PCA-based output compression is used. I think the paper would be stronger with either a principled derivation of this weighting or an ablation showing that the conclusions are robust to alternative weighting schemes.


In short, I think this is a promising paper with a strong empirical study and an appealing function-space viewpoint. The proposed metric is intuitive, the experiments are broad, and the regularization angle makes the work practically interesting. My main concern is that the theoretical and methodological foundations still feel somewhat preliminary relative to the scope of the claims: the central theorem is fairly limited, the metric depends on several design choices, and the connection between the fitted surrogate and a more intrinsic notion of simplicity is not yet fully established. Overall, I view this as a good paper that can be accepted, while its core conceptual claims would be stronger with tighter theory and more robustness analysis.

---

> ### Author Rebuttal · Authors · 2026-03-31
>
> We thank the reviewer for the positive assessment, careful reading, and thoughtful feedback. We agree with the reviewer’s central point that the theoretical and methodological foundations should more tightly match the scope of the claims, and we will revise the paper accordingly.
>
> **Theorem 3.1 vs. practical ED.**
>
> We agree that Theorem 3.1 mainly supports the _path-restriction step_, rather than fully justifying the practical ED estimator. The practical metric uses a coefficient-weighted effective degree because the exact algebraic degree is brittle and unsuitable for optimization. We can also provide a more direct rationale for ED itself: ED is a **stable functional** of the fitted coefficients, since its estimation error is directly controlled by the coefficient estimation error (e.g., via weighted $\ell_1$ or $\ell_2$ bounds). Under standard conditions where the surrogate coefficients are stable and consistent, the fitted ED estimator inherits the same stability. We will make this point explicit in the revision.
>
> **Basis dependence.**
>
> Our use of the **Chebyshev basis** is motivated by numerical stability and conditioning, rather than by a claim that the simplicity notion is intrinsic to that basis. To test this directly, we also evaluated **Legendre polynomials** on CIFAR-10 and found qualitatively similar results (see our response to **Reviewer 7X3t**), suggesting that the method is stable across orthogonal polynomial bases.
>
> **Hyperparameter choices.**
> The hyperparameters are not chosen independently for each experiment. Our protocol is to first sweep them on a simpler task to identify a stable operating range, and then transfer that surrogate pipeline to larger tasks, tuning mainly the regularization strength $\lambda$ (Sec. 6.5). In practice, CIFAR-10 suggests a useful default configuration: $n_p \approx B/2$ sampled pairs, resolution $r=4$, maximum degree $K=3$, randomized cosine sampling, and optional PCA. We will make this calibration protocol more explicit in the revision.
>
> **Linear weighting and scale effects.**
>
> We agree that the current draft does not sufficiently justify the specific linear weighting in Eq. (5). Our goal was to define a metric that is more robust than exact degree, monotone with respect to shifting spectral mass toward higher orders, and simple enough to serve as a regularizer. The choice is not unique: we also tested a _quadratic weighting_ variant, but found that it performs worse empirically. A plausible reason is that the current absolute-value weighting yields a _gradient_ that is less sensitive to _coefficient scale_.
>
> To address scale effects directly, our framework also includes a _normalized ED_ variant, which factors out overall coefficient magnitude. Empirically, however, the unnormalized ED often works better for regularization, likely because it naturally places more weight on output channels that matter more to the model’s prediction. We will clarify this distinction and report both normalized and unnormalized variants in the revision.
>
> **PCA dependence.**
>
> PCA in our method is an optional per-path output compression step used mainly for efficiency, rather than the source of the ED effect. To clarify this, we ran a small CIFAR-10 ablation with the efficiency-oriented setting, comparing no PCA against several PCA dimensions:
>
> | Output dimension after PCA | No PCA (original outputs) | 3D | 2D | 1D |
> |---:|:---:|:---:|:---:|:---:|
> | Test acc. | **89.99** | 89.84 | 89.90 | 88.75 |
>
> These results suggest that ED is not mainly driven by PCA: direct fitting on the original outputs already works well, PCA to 2–3 dimensions remains very close to no PCA, and only aggressive compression to 1D noticeably hurts performance.
>
> **When path-based surrogates preserve simplicity.**
>
> Beyond exact polynomial degree, we expect path-based polynomial surrogates to remain informative whenever the relevant notion of function-space simplicity is reflected in the path-restricted function by _concentration on low-order components_ or by good _low-degree approximation_. This includes exact degree as a special case, but also broader notions of "low complexity along sampled paths". However, if the relevant simpliciy notion is fundamentally non-polynomial after path restriction, the current ED surrogate may be less appropriate. We will clarify this scope in the revision.
>
> **Controlled study on PNNs.**
>
> We strongly agree that a controlled polynomial setting is valuable. We therefore conducted an additional experiment on **polynomial neural networks (PNNs)** to test whether ED recovers the intended complexity ordering in a setting where algebraic degree is intrinsic. The results show that ED indeed tracks the underlying degree order and possesses several useful properties (e.g., robustness against basis choice and PCA). Please see our response to **Reviewer 7X3t** for details.

---

> > ### Author Rebuttal · Reviewer_82s4 · 2026-03-31
> >
> > I appreciate the reviewers' detailed explanation and additional experiments. I think the results look good now and most of my concerns are resolved. I will maintain my score.

---

### Official Review · Reviewer_zM2W · 2026-03-09

**Soundness:** 3
**Presentation:** 3
**Significance:** 3
**Originality:** 3
**Overall Recommendation:** 4
**Confidence:** 3

**Summary:**

This paper proposes a function-space notion of simplicity for neural networks based on fitting univariate Chebyshev polynomial surrogates along data-dependent interpolation paths between examples.

**Compliance With Llm Reviewing Policy:**

Affirmed.

**Final Justification:**

The authors have addressed my concerns. I remain my positive score.

**Key Questions For Authors:**

1. Can you rerun the CIFAR-10 comparison in Table 1 with all hyperparameters selected on a validation split set only, and then report one final untouched test result per method?
2. When PCA is enabled in ED regularization, what exactly is differentiated? Is PCA recomputed per path and differentiated through, or treated as a stop-gradient/fixed projection?

**Strengths And Weaknesses:**

Strength:
1. The paper’s main strength is novelty. Combining path-based polynomial surrogates, a coefficient-weighted degree notion, and an end-to-end regularizer is conceptually interesting. The paper tries to stay in function space rather than parameter space, which is a meaningful departure from norm-style proxies.
2. The paper is clearly written and easy to get the high-level intuition.
3. The paper conducts a wide range of empirical studies including vision, contrastive learning, grokking, language tasks and RL.

A minor weakness:
1. The appendix ablation table is inconsistent. Table 5 says the experiment is for ResNet18, but the row is ViT-Tiny and the surrounding text also says all experiments are conducted on ResNet18. Figure 7 and Figure 8 captions describe the left-to-right panel order incorrectly.

---

> ### Author Rebuttal · Authors · 2026-03-31
>
> We thank the reviewer for the positive assessment and for highlighting the paper’s novelty, clarity, and broad empirical coverage. We also appreciate the reviewer's concrete and actionable comments on the evaluation protocol, implementation details, and presentation.
>
> **CIFAR-10 experiments with a separate validation split.**
>
> As requested, we reran the CIFAR-10 experiments under a protocol with a separate validation set and a final untouched test set. Concretely, we split the original 10k CIFAR-10 test set into 2k validation and 8k final test examples randomly. Hyperparameters were selected on the validation split only, and final accuracy was reported on the untouched 8k test split over 3 seeds.
>
> For the baselines, we searched over the same ranges as Appendix F.1. For ED, we adopted the efficiency-oriented configuration from Appendix F.1 / Table 6 and searched over $\lambda \in \lbrace1,2,\ldots,8\rbrace$.
>
> As shown in the table below, our main conclusions remain consistent under this protocol, and ED remains the best method compared with all baselines.
>
> | Setting                |     Baseline |        Mixup |          SAM |         ASAM |         Jacobian |         ED (ours) |
> | ---------------------- | -----------: | -----------: | -----------: | -----------: | ---------------: | -----------: |
> | Paper                  | 87.80 ± 1.17 | 88.83 ± 1.48 | 87.85 ± 1.27 | 87.85 ± 1.24 | 87.81 ± 0.17 | **90.18 ± 0.14** |
> | Rerun (val/test split) | 87.57 ± 0.46 | 88.32 ± 1.18 | 87.21 ± 0.39 | 87.94 ± 0.35 | 87.39 ± 0.72 | **89.79 ± 0.56** |
>
> **PCA implementation.**
>
> In our implementation, _PCA is performed separately for each sampled path_. For the outputs collected along a given path, we compute a path-specific PCA projection, project the path samples into the corresponding low-dimensional subspace, and then fit the polynomial surrogate in this reduced space. Gradients are computed starting from the **PCA outputs** and differentiated through the PCA decomposition process: after obtaining the path-specific projected representation, the ED regularization loss is computed and differentiated based on those projected coordinates. We will state this clearly in the revised paper.
>
> **Minor weakness.**
>
> Thank you for catching these issues. We appreciate the careful reading here. The model name in the caption of Table 5 and surrounding text should be ViT-Tiny, which we will fix in the revision of the paper. We will also fix the left-to-right panel order described in the captions of Figure 7 and Figure 8.

---

> > ### Author Rebuttal · Reviewer_zM2W · 2026-04-04
> >
> > The authors have addressed my concerns. I remain my positive score.

---

### Official Review · Reviewer_7X3t · 2026-03-12

**Soundness:** 4
**Presentation:** 4
**Significance:** 4
**Originality:** 4
**Overall Recommendation:** 5
**Confidence:** 4

**Summary:**

This study proposes a new proxy of the simplicity of functions (particularly neural networks).
The proposed approach is based on a linear blending of two samples with blending parameter $\alpha \in [0, 1]$. The output of the function $f(x(\alpha))$ for this linear blending $x(\alpha)$ is also governed by $\alpha$, and the effective degree of univariate polynomials approximating the path of $f(x(\alpha))$ for $\alpha \in [0, 1]$ is the proposed measure of simplicity. These three important requirements: generality across tasks and architectures, (ii) quantifiability at scale for trained models, and (iii) optimizabiltiy via differentiality. Experiments show that the proposed simplicity proxy better correlates with the generalization gap than sharpness and L2 norm. It also serves as a better signal of the start of the grokking. Besides, training with simplicity regularization improves performance on image and sentence classification.

**Compliance With Llm Reviewing Policy:**

Affirmed.

**Final Justification:**

This work has very solid contributions. The rebuttal addresses all three concerns with clear discussion and additional experiments.

**Key Questions For Authors:**

See Q1, Q2, and Q3 in the Weaknesses.

**Limitations:**

Yes.

**Strengths And Weaknesses:**

**Strength**

This work has the following strengths.
- This study presents an elegant idea of measuring the simplicity of functions.
- Approximating the interpolating path by a polynomial and using its effective dimension is a general and efficient approach. As the polynomial is univariate, the analysis can be done in one-dimensional space, and Theorem 3.1 provides a justification of this reduction.
- Experiments cover a wide range of tasks, optimizers, models, and learning strategies, which demonstrates the practicality and generality of the proposed metric.

---

**Weakness**

While I feel that this work is very solid, there are several suggestions and questions that could further improve this work.

First, there are several ad hoc choices of the method design, and I wonder if the authors tested other choices. Particularly, the linear blending in most cases is performed in the input space, but we could also try that in the embedding space or feature space at an intermediate layer. This allows one to see the simplicity of the successive network, and I wonder if this also brings some useful insights (-> Q1). The choice of the Chebysev function is also ad hoc. I understand the clearness of the effective dimension, but I wonder if other orthogonal bases (e.g., Fourier, Wavelet, etc.) also work, or if the Chebysev function (or polynomial bases) are particularly good for this application (-> Q2).

Second, the authors could try to apply the proposed simplicity metric to some simple neural networks for which (the upper bound of) the generalization gap is theoretically known. This may give a theoretical understanding of the relationship between the effective dimension and generalization gap (-> Q3).

I don't necessarily demand experiments for the abovementioned ideas but any clarification would help my understanding of this work.

---

> ### Author Rebuttal · Authors · 2026-03-31
>
> We thank the reviewer for the positive assessment and for the thoughtful suggestions. We are especially encouraged that the reviewer found the overall idea elegant and the empirical study broad and convincing.
>
> **Q1. Linear blending in intermediate feature space.**
> We agree this is an interesting direction. While the input-space interpolation serves as a architecture-agnostic default, our method is not restricted it. In fact, our BERT-based NLP experiments in Sec. 6.3 already operate in the _embedding space_ since direct interpolation over discrete tokens is not meaningful; we will further clarify this in the revision.
>
> To further address this point, we conducted an additional _feature-space interpolation_ experiment on CIFAR-10 by applying ED to paths constructed after the embedding layer and after the first Transformer attention block of ViT:
>
> | Interpolation layer | Baseline(w/o ED) | Input | Embedding | After block 1 |
> |---|:---:|:---:|:---:|:---:|
> | Test acc. | 86.49 | **89.99** | 88.92 | 88.18 |
>
> As shown above, ED still improves performance in intermediate feature spaces, although the gain is smaller than with input-space interpolation. A plausible reason is that intermediate-layer interpolation constrains the end-to-end function less directly, even though the regularization signal still backpropagates to earlier layers. That said, we note that our current implementation of feature-space interpolation is preliminary and believe it remains a promising direction to be further explored.
>
> **Q2. Basis choice and sampling strategy.**
> We agree that the basis choice deserves a clearer explanation. Our use of the **Chebyshev basis** is motivated mainly by numerical stability and conditioning in polynomial fitting, not by a claim that it is uniquely correct. More generally, the basis serves as a tractable surrogate representation of the path-restricted function. Since our simplicity notion is explicitly degree-based, **orthogonal polynomial bases** are especially natural; non-polynomial bases such as Fourier or wavelets would instead define related but conceptually different notions of spectral simplicity.
>
> To test basis dependence directly, we replaced the Chebyshev basis with the **Legendre basis** on CIFAR-10, using the same hyperparameters and three random seeds:
>
> | Basis | Test accuracy |
> |---|---:|
> | Chebyshev | 90.18 ± 0.14 |
> | Legendre | 89.89 ± 1.29 |
>
> The effect is small, suggesting that the method is stable across orthogonal polynomial bases rather than relying specifically on Chebyshev polynomials.
>
> Closely related to basis choice, we also performed a **sampling-strategy ablation** on CIFAR-10, comparing randomized cosine sampling (Eq. 8), fixed Chebyshev sampling (Eq. 7), and uniform sampling:
>
> | Resolution | 4 | 4 | 4 | 15 | 15 | 15 |
> |---|:---|:---|:---|:---|:---|:---|
> | Max degree | 3 | 3 | 3 | 14 | 14 | 14 |
> | Sampling | Randomized cosine | Fixed Chebyshev | Uniform | Randomized cosine | Fixed Chebyshev | Uniform |
> | Test acc. | **89.99** | 89.47 | 89.23 | 89.90 | **90.40** | 78.42 |
>
> These results support our sampling design: randomized cosine sampling works best in the low-resolution setting, while in the higher-degree setting uniform sampling becomes numerically unstable, and both Chebyshev-style schemes are much more reliable.
>
>
> **Q3. Connection to theoretically understood generalization settings.**
> We appreciate this suggestion. Part of this motivation is already reflected in Sec. 4.3, which discusses the connection between ED and more classical capacity-control notions such as Rademacher complexity; we will make this connection more explicit in the revision.
>
> To address the point more directly, we also add a controlled experiment on **polynomial neural networks (PNNs)**, where algebraic degree is intrinsic to the model class. We construct three target mappings (tasks 1-3) of increasing degree, together with _scaled versions_ of them (tasks 4-6) to test scale dependence, and train a simple 4-layer PNN with square activations to fit each task nearly perfectly. We then evaluate ED and its variants:
>
> | Task | Algebraic degree | ED (Chebyshev) | ED_norm (Chebyshev) | ED (Legendre) | ED (PCA-1, Chebyshev) | ED (PCA-2, Chebyshev) |
> |---|:---:|:---:|:---:|:---:|:---:|:---:|
> | task1 | 1 | 0.72 | 0.29 | 0.73 | 1.40 | 0.70 |
> | task2 | 2 | 1.17 | 0.78 | 1.33 | 2.12 | 1.34 |
> | task3 | 5 | 2.92 | 1.17 | 3.64 | 5.32 | 3.40 |
> | task4 | 1 | 1.34 | 0.28 | 1.35 | 2.67 | 1.34 |
> | task5 | 2 | 2.33 | 0.82 | 2.63 | 4.13 | 2.62 |
> | task6 | 5 | 5.30 | 1.17 | 6.64 | 9.62 | 6.16 |
>
> These controlled experiments show several useful properties of ED. First, ED preserves the **degree ordering** of the target mappings. Second, standard ED is not scale-invariant, while ED_norm is nearly invariant to output scaling. Third, the same ordering is preserved under the Legendre basis and under PCA-based output reduction, suggesting that the method is robust to basis choice and moderate dimensionality reduction.

---

> > ### Author Rebuttal · Reviewer_7X3t · 2026-04-02
> >
> > I appreciate the authors' clarification. The rebuttal addresses all three concerns with clear discussion and additional experiments. I'll keep my score.

---

### Official Review · Reviewer_PhFU · 2026-03-15

**Soundness:** 2
**Presentation:** 2
**Significance:** 3
**Originality:** 2
**Overall Recommendation:** 4
**Confidence:** 3

**Summary:**

The paper proposes effective degree as a simplicity measure which can also be used as a training regularizer. Broad empirical evidence is presented for its relevance to generalization.

**Compliance With Llm Reviewing Policy:**

Affirmed.

**Final Justification:**

The rebuttal addressed all of my concerns, and I increased my score accordingly. I would like to emphasize again that improving the paper's positioning would help.

**Key Questions For Authors:**

On top of the weaknesses listed above:
1. Can the authors better justify Eq. 5, the effective degree, beyond the shortcomings of algebraic degree?
Ideally, can they relate the effective degree to the degree of the neuromanifold, or to more classical notions such as Rademacher complexity or covering numbers?

2. Can the authors provide a clear positioning of the paper against the closest alternatives?
For example, a simple Venn diagram in the related work section would greatly help.

3. The proposed penalty (Eq. 13) looks like a degree-weighted $\ell_1$-penalty. Does it induce representation sparsity? What does this imply?

4. Can the authors include a failure analysis, along with ablations on sampling choices?

Overall, clear positioning of the paper, clear identification of the contributions ("We propose polynomial representations" is too broad), and focusing more on the properties of effective degree by *theoretically* showing its relevance as a simplicity metric would change my current scores on soundness, presentation and possibly originality depending on the degree of improvement.

**Limitations:**

On the theory side, the relevance of the proposed ED to simplicity should be discussed. On the empirical side, the paper should discuss when the method breaks and the effect of sampling choices.

**Strengths And Weaknesses:**

**Strengths**
The empirical evidence strongly supports the method. Although the baselines are limited, they seem well chosen. The overall direction of the paper is promising. In that sense, the work has reasonable significance, even if I am less convinced by the level of novelty and technical support.

**Readability**
I suggest a significant focus on positioning the paper. The related work section should be used to identify the differences from the closest alternatives, so that the paper becomes more understandable and its contributions become more clear. In particular, given the title of the work and contributions 1–2, the paper omits several recent and directly relevant lines: 1) PNNs: polynomial networks and neuromanifold geometry; 2) neuroalgebraic geometry (e.g. Shahverdi et al.): linking dimension, degree, and singularities of neural function spaces to learning; 3) polynomial KANs, especially NMR-KAN which focuses on degree-based functional simplicity control. These lines are directly relevant to the paper’s claims about polynomial representations and the relation between representation degree and functional capacity. Clearly stating the architectural differences from these works, and consequently the notion of simplicity being used, would make the paper’s positioning more clear. Also, although the baselines are limited, they seem well chosen, and I would suggest aligning/focusing the related work section with the compared methods.

**Soundness**
There is a mismatch between the complexity (in the sense that it has multiple steps) of the proposed pipeline and what is theoretically shown. Theorem 3.1 supports only a part of the overall work. Equation 5, the effective degree, is surprisingly left unmotivated beyond the shortcomings of algebraic degree. "Why this definition rather than another one" is unclear. *This is particularly important since effective degree seems to be the main novelty of the work.* The proposed penalty (Eq. 13) also looks like a degree-weighted $\ell_1$-penalty term (in view of Eq. 10), but it is unclear whether it induces representation sparsity. The paper should also clarify how the ridge regression in Eq. 12 is solved in practice, e.g. via QR, conjugate gradient, or another iterative solver. If I understood the methodology correctly, I assume the memory should not be a bottleneck for the proposed method?

**Originality**
The empirical angle is useful, but the originality is harder to assess positively because the paper is not well positioned against the literature. As written, it is not sufficiently clear what is genuinely new relative to the closest alternatives.

**Significance**
The problem setting is meaningful, and the empirical evidence suggests the method may be useful in practice. However, the paper would be stronger if it included a more clear failure analysis and ablations on sampling choices (e.g. how much is gained by Eq. 8 over uniform sampling?).

---

> ### Author Rebuttal · Authors · 2026-03-31
>
> We thank the reviewer for the careful and constructive feedback.
>
> **Positioning and contributions.**
> Thank you for pointing us to related work on PNNs, neuroalgebraic geometry, and KANs. We will revise the related work section to position our paper more clearly. The main distinction is that these lines study **specific model classes** whose functions are (semi-)polynomial, whereas our goal is to define a **general simplicity metric** for arbitrary neural architectures rather than propose or regularize a particular polynomial architecture. Our polynomial representation and effective degree (ED) are therefore **architecture-agnostic** and applicable to modern models such as Transformers.
>
> Technically, our method also differs from neuroalgebraic geometry in scope: rather than characterizing the **global geometry** of a model family, we use a **local, low-dimensional surrogate** of the path-restricted function, which makes the resulting simplicity metric computationally efficient and differentiable. We agree that the current wording around "polynomial representations" is too broad, and we will revise the manuscript to make the contribution more precise.
>
> **Theoretical justification of ED.**
> We agree that Theorem 3.1 does not fully justify the entire practical pipeline. Its intended role is narrower: it supports the _path-restriction idea_ in the polynomial setting. The practical ED metric is introduced as a stable surrogate for exact algebraic degree, motivated by numerical robustness and differentiability. The specific choice is not unique; we also tested alternative variants and found that the current definition performs best empirically. We posit that this is due to the absolute-value weighting naturally introducing a desirable **scale-invariance** of the gradient with respect to coefficient magnitude.
>
> We also note that ED can be related to more classical complexity notions: Sec. 4.3 already discusses its connection to _Rademacher-style capacity control_ and generalization bounds, and we will make this connection more explicit in the revision. Regarding _neuromanifold degree_, we believe a clean general connection is difficult: in the current neuroalgebraic-geometry literature, neuromanifold degree is typically studied in **algebraic settings** where the functions are (semi-)polynomial. For more general model classes, this notion is not always available in the same sense. For polynomial cases such as PNNs, however, our metric can recover the intended degree ordering; please see our response to **Reviewer 7X3t**.
>
> **Connection to sparsity.**
> We agree that the current wording around Eq. (13) is imprecise. A more accurate interpretation is that Eq. (13) is a _degree-weighted coefficient penalty_ that shifts spectral mass toward lower orders. It therefore does not necessarily induce exact sparsity in the classical sense (e.g., a representation focusing solely on higher orders is sparse yet high degree).
>
> **Implementation details.**
> In practice, Eq. (12) is solved using `torch.linalg.solve` in PyTorch, which relies on _LU-based_ linear solves. The overhead is modest because each fit is only a small _per-path linear solve_; in practice, the main cost comes from repeating this operation across many sampled paths rather than storing a large global object.
>
> **Failure analysis and sampling choices.**
> Thank you for raising this point. We include a failure case to clarify the limitation of ED regularization, and provide sampling-choice ablations in our response to **Reviewer 7X3t**.
>
> Motivated by prior work on simplicity bias, we hypothesize that ED regularization may fail when the simpler feature is also the less robust one. To test this, we construct _MNIST-CIFAR_ as in [1], a synthetic dataset formed by concatenating MNIST (0 vs. 1) and CIFAR (automobile vs. truck), where the task can be solved using either simple MNIST features or more complex CIFAR features. We train ViT-Tiny with standard training and with ED regularization:
>
> | Metric | Baseline | ED |
> |---|---:|---:|
> | Test acc. | 99.85 | 99.90 |
> | Test acc. (MNIST randomized) | 48.05 | 47.95 |
> | Test acc. (CIFAR randomized) | 99.85 | 99.90 |
>
> Both models achieve near-perfect standard accuracy, but randomizing the MNIST part causes accuracy to collapse, while randomizing CIFAR has almost no effect. This shows that both models rely almost entirely on the simpler MNIST signal. ED regularization does not reduce this reliance or improve robustness in this setting. We view this as a plausible failure case of ED: when simple features are easier to exploit but less desirable for robust generalization, ED may fail to encourage the use of more generalizable complex features.
>
> **Reference:**
>
> [1] Shah et al., The Pitfalls of Simplicity Bias in Neural Networks, NeurIPS 2020.

---

> > ### Author Rebuttal · Reviewer_PhFU · 2026-04-04
> >
> > I thank the authors for their rebuttal. The rebuttal addressed all of my concerns, and I increased my score accordingly. Now that the scope of the paper is more clear to me, I believe the authors need not discuss all of the papers I mentioned in my initial review. However, I would like to emphasize again that improving the paper's positioning would help.

---

### Decision · Program_Chairs · 2026-04-30

**Decision:**

Accept (regular)

**Comment:**

The paper’s core idea is to define a function-space simplicity metric for neural networks using polynomial surrogates along interpolation paths between data points. The authors fit a low-dimensional univariate polynomial representation of the model’s predictions along these paths, then measure simplicity using an effective degree (ED) derived from the fitted coefficients. They argue that ED is both a useful post-hoc predictor of generalization and a differentiable regularizer that can improve training. Empirically, they report that ED predicts generalization better than baselines like sharpness and can improve performance in image classification, text classification, CLIP fine-tuning, grokking settings, and reinforcement learning.

All reviewers agreed to accept this paper based on grounds including (1) broad empirical scope, (2) the function-space perspective is interesting, (3) the proposed metric is both measurable and optimisable.